# 🐱 MIO: A Foundation Model on Multimodal Tokens

## ABSTRACT

In this paper, we introduce MIO, a novel foundation model built on multimodal tokens, capable of understanding and generating speech, text, images, and videos in an end-to-end, autoregressive manner. While the emergence of large language models (LLMs) and multimodal large language models (MM-LLMs) propels advancements in artificial general intelligence through their versatile capabilities, they still lack true any-to-any understanding and generation. Recently, the release of GPT-4o has showcased the remarkable potential of any-to-any LLMs for complex real-world tasks, enabling omnidirectional input and output across images, speech, and text. However, it is closed-source and does not support the generation of multimodal interleaved sequences. To address this gap, we present MIO, which is trained on a mixture of discrete tokens across four modalities using causal multimodal modeling. MIO undergoes a four-stage training process: (1) alignment pre-training, (2) interleaved pre-training, (3) speech-enhanced pre-training, and (4) comprehensive supervised fine-tuning on diverse textual, visual, and speech tasks. Our experimental results indicate that MIO exhibits competitive, and in some cases superior, performance compared to previous dual-modal baselines, any-to-any model baselines, and even modality-specific baselines. Moreover, MIO demonstrates advanced capabilities inherent to its any-to-any feature, such as interleaved video-text generation, chain-of-visual-thought reasoning, visual guideline generation, instructional image editing, etc. Anonymous codes and supplemental materials are available at https://anonymous.4open.science/r/anonymous_MIO-DDE5.

## 1 INTRODUCTION

The advent of Large Language Models (LLMs) is commonly considered the dawn of artificial general intelligence (AGI) (OpenAI et al., 2023; Bubeck et al., 2023), given their generalist capabilities such as complex reasoning (Wei et al., 2022), role playing (Wang et al., 2023c), and creative writing (Wang et al., 2024a). However, original LLMs lack multimodal understanding capabilities. Consequently, numerous multimodal LLMs (MM-LLMs) have been proposed, allowing LLMs to understand images (Li et al., 2023b; Alayrac et al., 2022), audio (Borsos et al., 2023; Rubenstein et al., 2023; Tang et al., 2023; Das et al., 2024), and other modalities (Lyu et al., 2023; Zhang et al., 2023d; Moon et al., 2023). These MM-LLMs typically involve an external multimodal encoder, such as EVA-CLIP (Sun et al., 2023b) or CLAP (Elizalde et al., 2022), with an alignment module such as Q-Former (Li et al., 2023b) or MLP (Liu et al., 2023b) for multimodal understanding. These modules align non-textual-modality data features into the embedding space of the LLM backbone.

Another line of work involves building **any-to-any** and end-to-end MM-LLMs that can input and output non-textual modality data. Typically, there are four approaches: (1) Discrete-In-Discrete-Out (DIDO): Non-textual modality data is discretized using vector quantization techniques (van den Oord et al., 2017; Esser et al., 2020) and then fed into LLMs (Ge et al., 2023b; Zhan et al., 2024; Liu et al., 2024). (2) Continuous-In-Discrete-Out (CIDO): The LLM backbones intake densely encoded non-textual modality data features and generate their quantized representations (Diao et al., 2023; Team et al., 2023). (3) Continuous-In-Continuous-Out (CICO): The LLMs both understand and generate non-textual modality data in their densely encoded representations (Sun et al., 2023c;a; Dong et al., 2023; Zheng et al., 2023; Wu et al., 2023). (4) Autoregression + Diffusion (AR + Diff): The autoregressive and diffusion modeling are integrated in a unified LLM (Zhou et al., 2024; Xie

Table 1: The comparison between previous models and MIO (ours). **I/O Consistency** indicates whether the model ensures that the input and output representations for the same data remain consistent. **Uni. Bi. SFT** refers to whether the model undergoes a unified (Uni.) supervised fine-tuning (SFT) for both multimodal understanding and generation (Bi.=Bidirectional). **Multi-Task SFT** assesses whether the model undergoes a comprehensive SFT that includes diverse tasks, with at least visual question answering tasks. **MM. Inter. Output** evaluates whether the model supports the generation of multimodal interleaved (MM. Inter.) sequences. We refer readers to §1 for the definitions of the different modeling approaches.

| Models | Emu1 (Sun et al., 2023c) | Emu2 (Sun et al., 2023a) | SEED-LLaMA (Ge et al., 2023b) | AnyGPT (Zhan et al., 2024) | CM3Leon (Yu et al., 2023), Chameleon (Team, 2024) | Gemini (Reid et al., 2024) | Transfusion (Zhou et al., 2024) | MIO (ours) |
|---|---|---|---|---|---|---|---|---|
| I/O Consistency | ✗ | ✓ | ✓ | ✓ | ✓ | ✗ | ✗ | ✓ |
| Uni. Bi. SFT | ✗ | ✗ | ✓ | ✓ | ✓ | ✓ | ✗ | ✓ |
| Multi-Task SFT | ✓ | ✓ | ✓ | ✗ | ✓ | ✓ | ✗ | ✓ |
| Speech I/O | ✗/✗ | ✗/✗ | ✗/✗ | ✓/✓ | ✗/✗ | ✓/✗ | ✗ | ✓/✓ |
| Video I/O | ✓/✓ | ✓/✓ | ✓/✓ | ✗/✗ | ✗/✗ | ✓/✗ | ✗ | ✓/✓ |
| Voice Output | ✗ | ✗ | ✗ | ✗ | ✗ | ✗ | ✗ | ✓ |
| MM. Inter. Output | ✗ | ✗ | ✓ | ✗ | ✗ | ✗ | ✗ | ✓ |
| Modeling | CICO | CICO | DIDO | DIDO | DIDO | CIDO | AR+Diff | DIDO |

et al., 2024; Li et al., 2024b). Although these works have succeeded in building MM-LLMs unifying understanding and generation, they exhibit some drawbacks, as illustrated in Table 1. For example, Emu1 (Sun et al., 2023c) and Emu2 (Sun et al., 2023a) explore the autoregressive modeling of three modalities: text, images, and videos. SEED-LLaMA (Ge et al., 2023b) proposes a new image quantizer aligned with LLMs' embedding space and trains the MM-LLMs on images and videos. However, neither considers the speech modality, which is heterogeneous from visual modalities like videos and images. Although AnyGPT (Zhan et al., 2024) has explored settings involving four modalities, including text, image, speech, and music, it lacks video-related abilities, voice synthesis, and comprehensive multi-task supervised fine-tuning, leading to limited multimodal instruction-following and reasoning capabilities. Furthermore, AR + Diff approaches, such as Transfusion (Zhou et al., 2024), suffer from limited multimodal understanding capabilities because the multimodal inputs are noised for denoising modeling, and the image tokenizer used (*i.e.*, VAE (Kingma & Welling, 2013)) is suitable for image generation rather than image understanding.

Moreover, most of current MM-LLMs are typically dual-modal, combining text with another modality, such as images. Although previous works, such as Meta-Transformer (Zhang et al., 2023d) and Unified-IO 2 (Lu et al., 2023), have explored omni-multimodal understanding settings with more than two non-textual modalities, they still lag significantly behind their dual-modal counterparts, especially in terms of multimodal instruction-following capabilities. Moreover, these MM-LLMs are typically focused on understanding only, neglecting the important aspect of multimodal generation. Several works have enabled LLMs to call external tools to address this issue. For example, HuggingGPT (Shen et al., 2023) generates textual image descriptions for external diffusion models to synthesize images. GPT-4 (OpenAI et al., 2023) can utilize either an image generator like DALL-E 3 (Betker et al., 2024) or a text-to-speech (TTS) tool like Whisper (Radford et al., 2022) to support multimodal generation.[1] However, these methods are not end-to-end, relying on the text modality as an interface.

Recently, the release of GPT-4o has demonstrated the capabilities of any-to-any and end-to-end foundation models.[2] It is the first foundational model to accept multimodal tokens as inputs and generate multimodal tokens within a unified model while also demonstrating strong abilities in complex multimodal instruction-following, reasoning, planning, and other generalist capabilities. Furthermore, as the continuous scaling up of LLMs in the community depletes high-quality language tokens, GPT-4o verifies a new source of data for LLM training: multimodal tokens. This approach suggests that the next generation AGI could derive more knowledge from multimodal tokens when language tokens are exhausted. However, GPT-4o is closed source and focuses primarily on end-to-end support for speech I/O, image I/O, 3D generation, and video understanding. Its recent open-source "alternatives", such as VITA (Fu et al., 2024), still lack the ability to *generate* data of all supported modalities, particularly for the generation of multimodal interleaved sequences.

---

[1] https://openai.com/index/chatgpt-can-now-see-hear-and-speak/
[2] https://openai.com/index/hello-gpt-4o/

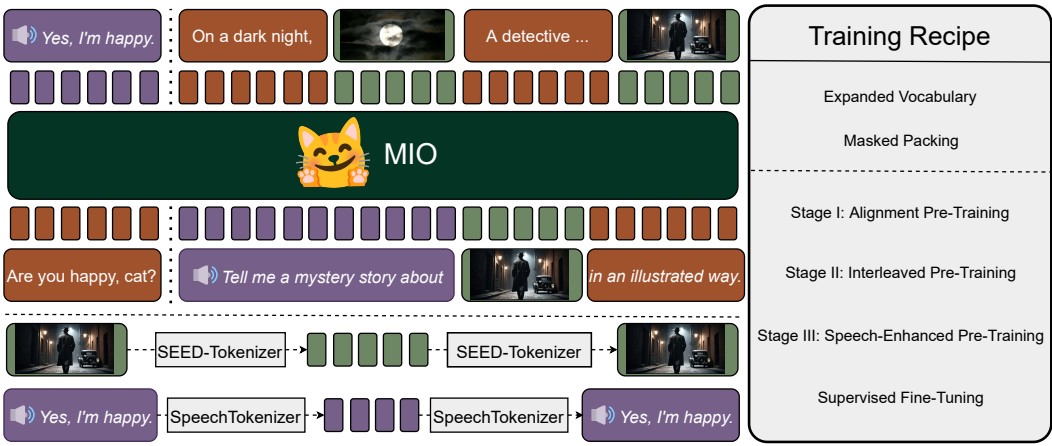

Figure 1: The framework of MIO and its training recipe.

To address the aforementioned issues, we introduce MIO (Multimodal Input and Output, or Multimodal Interleaved Output), the first open-source any-to-any foundation model that unifies multimodal understanding and generation across four modalities–text, image, speech (with voice), and video, while enabling the generation of multimodal interleaved sequences. Specifically, MIO is built on discrete multimodal tokens that capture both semantic representations through contrastive loss and low-level features via reconstruction loss (Ge et al., 2023a; Zhang et al., 2023b) from raw multimodal data. Due to the consistent data format shared with textual corpora, the model can treat non-textual modalities as "foreign languages", allowing it to be trained with the next-token-prediction. Note that since the representation of an image remains the same whether it is used as an input or an output, our model flexibly supports multimodal interleaved sequence generation, where an image functions simultaneously for both understanding and generation. Moreover, we employ three-stage pre-training with an additional SFT stage to effectively train the model for modality scaling.

Our experimental results show that MIO, trained on a mixture of four modalities, demonstrates competitive performance compared to its dual-modal counterparts and previous any-to-any multimodal language model baselines. Additionally, MIO is the first model to demonstrate interleaved video-text generation, chain-of-visual-thought reasoning, and other emergent abilities relying on any-to-any and multimodal interleaved output features (*c.f.*,§3.5).

## 2 METHOD

Firstly, we elaborate on our modeling approach, which supports multimodal token input and output, as well as causal language modeling (CausalLM), in §2.1. Secondly, we describe our three-stage pre-training procedures in §2.2. Thirdly, we provide details of our comprehensive supervised fine-tuning on diverse multimodal understanding and generation tasks in §2.3.

### 2.1 MODELING

As illustrated in Figure 1, the framework of MIO involves three parts: (1) multimodal tokenization, (2) causal multimodal modeling, and (3) multimodal de-tokenization.

**Multimodal Tokenization.** In our work, we use SEED-Tokenizer (Ge et al., 2023a) as our image tokenizer and SpeechTokenizer (Zhang et al., 2023b) as our speech tokenizer. SEED-Tokenizer encodes images using a ViT (Dosovitskiy et al., 2021) derived from BLIP-2 (Li et al., 2023b), and then converts the encoded features into fewer tokens with causal semantics via Q-Former (Li et al., 2023b). These features are subsequently quantized into discrete tokens that are well-aligned with the language model backbone's textual space. The codebook size for these discrete image tokens is 8192. SEED-Tokenizer transforms each image into a 224x224 resolution and quantizes it into 32

tokens. We use two special tokens, <IMAGE> and </IMAGE>, to indicate the start and end of the image tokens per image, respectively.

As for videos, we first apply specific frame-cutting methods to convert videos into image sequences. In our training data processing procedures, the number of frames for each video is dynamically determined by its duration, the length of its context, or its scene switching[3] to (1) avoid exceeding the LLM backbone's context window limit, and (2) capture complete but concise information of the video. Each frame is then tokenized in the same manner as an image.

In terms of speech, SpeechTokenizer (Zhang et al., 2023b) leverages an 8-layer RVQ (Lee et al., 2022) to tokenize speech into tokens with 8 codebooks, with each codebook derived from one layer. Since the first layer's quantization output is distilled from HuBERT (Hsu et al., 2021), which encodes more semantic information, SpeechTokenizer can separate content tokens and timbre tokens from a quantized speech. The first-layer quantization is treated as content quantization, while the remaining layers' quantization is treated as timbre quantization. SpeechTokenizer encodes speech into 50 tokens per second for each codebook, resulting in 400 tokens per second with all eight codebooks. To improve context efficiency, we drop the last four layers' codebooks and only use the content codebook and the first three timbre codebooks. Our vocabulary size for the speech modality is $1024 \times 4 = 4096$.

Since the open-source pretraining-level speech data is collected from individuals with diverse voices, the timbre tokens exhibit a relatively random and noisy pattern, while the content tokens are more fixed-pattern and better aligned with the corresponding transcriptions. Given these priors in speech tokens, it is important to choose the proper interleaving mode of speech tokens (Copet et al., 2023). We denote the four codebooks as $\mathcal{A}$, $\mathcal{B}$, $\mathcal{C}$, and $\mathcal{D}$, where $\mathcal{A}$ is the codebook for content tokens and the remaining three are for timbre tokens. For simplicity, assuming that we have only two tokens for each codebook in a tokenized speech sequence (i.e., $a_1a_2$, $b_1b_2$, $c_1c_2$, and $d_1d_2$), there are two interleaving patterns for causal multimodal modeling: (1) sequential interleaving pattern: $a_1a_2b_1b_2c_1c_2d_1d_2$ and (2) alternating interleaving pattern: $a_1b_1c_1d_1a_2b_2c_2d_2$.

In our preliminary experiments, we observed that text-to-speech generation (TTS) training is difficult to converge when using the alternating interleaving pattern because the noisy and random timbre tokens ($b_1c_1d_1$) tend to mislead the continuations. Moreover, the speech-to-text understanding (ASR) performance improves much more slowly during training with the alternating interleaving pattern due to the sparsity of semantic information in the timbre tokens. As a result, we drop the timbre tokens for speech understanding and use the sequential interleaving pattern for speech generation. We use <SPCH> and </SPCH> as special tokens to indicate the start and end of the speech token sequence.

**Causal Multimodal Modeling.** As illustrated in Figure 1, the speech and images, including video frames, are tokenized by SpeechTokenizer (Zhang et al., 2023b) and SEED-Tokenizer (Ge et al., 2023a), respectively. We add the 4096 speech tokens and 8192 image tokens to the LLM's vocabulary. In addition, we introduce four new special tokens, namely <IMAGE>, </IMAGE>, <SPCH>, and </SPCH>, to the vocabulary. Consequently, the embedding layer of the LLM backbone and the language modeling head are extended by $4096 + 8192 + 4 = 12292$ to support the embedding and generation of these new tokens. The image tokens contain *causal* semantics due to the use of a *Causal* Q-Former (Ge et al., 2023a), and the speech tokens are intrinsically causal due to their temporal nature. Therefore, these multimodal tokens are as suitable for autoregressive training as textual tokens, allowing us to unify the training objectives for understanding and generation of multimodal tokens into next-token-prediction with cross-entropy loss. The training objective is thus:

$$\mathcal{L} = -\sum_{t=1}^{T} \log P(x_t \mid x_{<t}; \theta) \tag{1}$$

where $x_t$ represents the discrete multimodal tokens, and $\theta$ denotes the parameters of the LLM backbone. We use the pre-trained model, Yi-6B-Base (AI et al., 2024), for initialization.

Furthermore, to eliminate the computational inefficiency caused by <PAD> tokens, we use the masked packing strategy (Lu et al., 2023; Liu et al., 2024; Dehghani et al., 2023). Specifically, the samples are concatenated along the sequence length dimension until the context window is full. Then, we

---

[3]https://github.com/Breakthrough/PySceneDetect

construct the causal attention mask for the tokens of each sample and mask out all the tokens of the other samples.

**Multimodal De-Tokenization.** After the generation of multimodal tokens, it is essential to use modality-specific decoders to reconstruct the images or speech from the codes. Specifically, for image tokens, we directly utilize SEED-Tokenizer's decoder, which involves an MLP projection to convert the discrete codes into dense latents. These latents condition an off-the-shelf diffusion model (Rombach et al., 2022) to generate the images in the pixel space (Ge et al., 2023a). The vanilla SpeechTokenizer (Zhang et al., 2023b) involves generating timbre tokens through a *non-autoregressive* model outside the language model, and then feeding the concatenated content and timbre tokens into the SpeechTokenizer decoder to synthesize speech. In our work, to inject the timbre priors into the multimodal language model itself, the timbre tokens are also generated by the *autoregressive* language model.

## 2.2 Pre-Training

As shown in Table 2, we use a three-stage strategy for pre-training, with each stage targeting different objectives. The three stages are: (1) Alignment Pre-training: This stage focuses on learning a multimodal representation more aligned with the language space. (2) Interleaved Pre-training: This stage aims to obtain a multimodal representation with richer contextual semantics. (3) Speech-enhanced Pre-training: This stage specifically enhances the model's speech-related capabilities, while concurrently replaying data from other modalities. For more details on the pre-training data and its processing procedures, we refer the readers to Appendix A.

Table 2: Pre-training stages and their details. We use "Inter" to denote "Interleaved" for short. We provide batch sizes for each data type per GPU in image-text pair data:language-only data:(image-text interleaved data + video data):speech-text pair data. See Appendix A and Appendix B for more details including pre-training data sources, data cleaning procedures, pre-training hyperparameters, etc.

| Pre-training Stage Objective | Stage I Multimodal Alignment | Stage II Multimodal Interleaving | Stage III Speech Enhancement |
|---|---|---|---|
| **Image-Text Pair** | SBU, CC3M, LAION-COCO, JourneyDB | SBU, CC3M, LAION-COCO, JourneyDB | CC3M LAION-COCO |
| **Language-Only** | RefinedWeb | RefinedWeb | RefinedWeb |
| **Image-Text Inter** | - | OBELICS, MMC4-core-ff | MMC4-core-ff |
| **Video-Text Pair** | - | WebVid-10M | WebVid-10M |
| **Video-Text Inter** | - | HowTo-100M, YT-Temporal-180M | HowTo-100M, YT-Temporal-180M |
| **Speech-Text Pair** | Libriheavy | Libriheavy | Libriheavy |
| **GPUs** **Training Steps** **Batch Size** | 128 A800-80GB 24,800 12:2:0:2 | 128 A800-80GB 12,800 2:2:6:6 | 8 A800-80GB 32,200 2:1:1:12 |

**Stage I: Alignment Pre-Training.** To fully leverage the superior capabilities of the pre-trained LLM backbone, it is essential to align the non-textual modality data representations with text. There are two types of pre-training data for image-text multimodal learning: (1) Image-text paired data: This data has well-aligned dependencies between images and text. (2) Image-text interleaved data: This data features more natural and contextual dependencies but is less aligned. Note that in our setting, video-text paired and interleaved data can be treated as image-text interleaved data, with videos being sequential images interleaved with text. Therefore, in this stage, we exclude the image-text interleaved data and video data to ensure the most aligned pattern between images and text.

**Stage II: Interleaved Pre-Training.** In this stage, we extend the data used for pre-training to include image-text interleaved data (including video-text data) as a novel image-text dependency

pattern. The image-text interleaving pattern has a different nature compared to pairing patterns. Although Li et al. (2023b) and Sun et al. (2023c) argued that interleaved image-text data mainly serves for *multimodal in-context learning*, we argue that it is also essential for context-aware image generation where images are generated based on specific context, rather than a precise description of the image content. For example, in image-text interleaved data, the text might serve as the image's preceding or continuing context, rather than its description. This pattern significantly differs from the previous descriptive image generation demonstrated in image-text paired data, where images are generated based on precise and detailed text that clearly describe the content of the images (Team et al., 2023). Therefore, context-aware image generation is essential for tasks such as *chain-of-visual-thought reasoning* or *visual storytelling* (Team et al., 2023; Huang et al., 2016), where images are generated without textual descriptions. Due to the lack of benchmarks and evaluation metrics for context-aware image generation, we provide some demonstrations in §3.5 to showcase the potential of our model in visual storytelling, interleaved video-text generation, instructional image editing, chain-of-visual-thought reasoning, multimodal in-context learning, etc.

Moreover, in this stage, due to the extensive training on image-text paired data in Stage I, we can reduce its mixing ratio to the minimal essential scale for replay to avoid catastrophic forgetting. This allows us to increase the batch size for image-text interleaved data, video data, and speech data.

**Stage III: Speech-Enhanced Pre-Training.** The speech tokenizer that we use generates 200 tokens for each second of audio. Given that the duration of a speech sample can be 15 seconds, this results in around 3,000 tokens per sample. In comparison, the image tokenizer produces only 32 tokens per image. This creates a significant disparity in the number of tokens among different modalities. Consequently, our training data is dominated by speech tokens. If we mix all the different modalities according to their original proportions for training, the model would likely become overly focused on speech, at the expense of other modalities.

To address this issue, we implement a three-stage strategy that gradually increases the proportion of speech tokens. In Stage I, speech-text data accounts for 12.5% of the training tokens, which rises to 37.5% in Stage II, and finally reaches 75.0% in Stage III. This incremental increase in the proportion of speech tokens ensures that the model's performance in non-speech modalities is not compromised by the speech modality, while also allowing for the optimization of the model's speech capabilities.

Furthermore, we keep the data mixing ratio for other modalities of pre-training data at the minimal essential scales for replay, and we only use the high-quality subsets of them in this stage. This stage requires significantly fewer compute resources, due to the foundation laid in the previous stages.

We refer the reader to Appendix B for more details about the hyperparameters and prompt templates.

## 2.3 SUPERVISED FINE-TUNING

As shown in Table 9, our model undergoes comprehensive and systematic supervised fine-tuning (SFT) with 16 different tasks and 34 diverse open-source datasets. The chat template used for SFT is the same as that used for Yi-6B-Chat (AI et al., 2024), and only the assistant responses are supervised. We refer the reader to Appendix C for more details about the hyperparameters and prompt templates.

## 3 EXPERIMENTS

In this section, we present our quantitative evaluation results across various domains: image-related tasks (§3.1), speech-related tasks (§3.2), and video-related tasks (§3.3). Due to the lack of benchmarks for several advanced and emergent abilities of any-to-any multimodal LLMs, we also provide numerous qualitative demonstrations (§3.5) to demonstrate these capabilities. We refer the reader to Appendix D for more details, including the decoding hyperparameters and prompt templates.

### 3.1 IMAGE-RELATED TASKS

**Image Understanding.** We compare our models with Emu (Sun et al., 2023c), SEED-LLaMA (Ge et al., 2023b), AnyGPT (Zhan et al., 2024), Flamingo (Alayrac et al., 2022), Kosmos-1 (Huang et al., 2023), MetaLM (Hao et al., 2022), IDEFICS (Laurençon et al., 2023), CM3Leon (Yu et al., 2023), InstructBLIP (Dai et al., 2023), Qwen-VL-Chat (Bai et al., 2023), and LLaVA 1.5 (Liu

Table 3: Experimental results for image understanding abilities. "Imagen" denotes whether the model is capable of generating images. "Speech" denotes whether the model supports speech modality. "I" denotes the instruction tuned version. The metrics used are CIDEr for COCO, MCQ accuracy for the SEED Bench, and VQA accuracy for the other tasks, following the standard procedures. In all cases, higher scores indicate better performance.

| Models | Imagen | Speech | COCO | VQAv2 | OKVQA | VizWiz | SEED Bench |
|---|---|---|---|---|---|---|---|
| Emu-Base (14B) | ✓ | ✗ | 112.4 | 52.0 | 38.2 | 34.2 | 47.3 |
| Emu-I (14B) | ✗ | ✗ | 120.4 | 57.2 | 43.4 | 32.2 | 58.0 |
| SEED-LLaMA-I (8B) | ✓ | ✗ | 124.5 | 66.2 | 45.9 | 55.1 | 51.5 |
| AnyGPT (8B) | ✓ | ✓ | 107.5 | - | - | - | - |
| Flamingo (9B) | ✗ | ✗ | 79.4 | 51.8 | 44.7 | 28.8 | 42.7 |
| Flamingo (80B) | ✗ | ✗ | 84.3 | 56.3 | 31.6 | | - |
| Kosmos-1 (1.6B) | ✗ | ✗ | 84.7 | 51.0 | - | 29.2 | - |
| MetaLM (1.7B) | ✗ | ✗ | 82.2 | 41.1 | 11.4 | - | - |
| IDEFICS-I (80B) | ✗ | ✗ | 117.2 | 37.4 | 36.9 | 26.2 | 53.2 |
| CM3Leon (7B) | ✓ | ✗ | 61.6 | 47.6 | 23.8 | 37.6 | - |
| InstructBLIP (8.1B) | ✗ | ✗ | - | - | - | 34.5 | 58.8 |
| Qwen-VL-Chat (13B) | ✗ | ✗ | - | 78.2 | 56.6 | 38.9 | 58.2 |
| LLaVA 1.5 (7B) | ✗ | ✗ | - | 78.5 | - | 50.0 | 58.6 |
| MIO-Instruct (7B) | ✓ | ✓ | 120.4 | 65.5 | 39.9 | 53.5 | 54.4 |

et al., 2023a). We evaluate our models in diverse tasks, including: (1) image captioning on MS-COCO (Lin et al., 2014) Karpathy test split with CIDEr score (Vedantam et al., 2014) as the metric, (2) three visual question-answering benchmarks, *i.e.*, VQAv2 (Goyal et al., 2016) (test-dev split), OK-VQA (Marino et al., 2019) (val split), and VizWiz (Gurari et al., 2018), with VQA accuracy as the metric, and (3) SEED-Bench (Li et al., 2023a), a comprehensive visual question-answering benchmark including 9 dimensions with MCQ accuracy as the metric. The scores for all baselines are copied from their reports. As shown in Table 3, our MIO-Instruct is ranked in the top group among all baselines, demonstrating its competitive image understanding performance. Although SEED-LLaMA achieved better scores compared to our model, we additionally support the speech modality. It is also noteworthy that MIO, with a size of approximately 7 billion parameters, outperforms several larger models such as Emu-14B and even IDEFICS-80B.

**Image Generation.** We compare our models with Emu (Sun et al., 2023c), SEED-LLaMA (Ge et al., 2023b), GILL (Koh et al., 2023), and AnyGPT (Zhan et al., 2024) for image generation. We use two benchmarks, *i.e.*, MS-COCO (Lin et al., 2014) Karpathy test split and Flickr30K (Plummer et al., 2015). Following GILL (Koh et al., 2023) and SEED-LLaMA (Ge et al., 2023b), we use CLIP-I as the metric that evaluates the similarity between the generated images and the ground-truth images with the image encoder in CLIP (Radford et al., 2021). As shown in Table 4 and Table 12 the pre-trained model and instruction-tuned model of MIO both have com-

Table 4: Image generation evaluation by CLIP-I score. "I" denotes the instruction tuned version. Higher values are better.

| Models | MS-COCO | Flickr30K |
|---|---|---|
| Emu-Base | 66.46 | 64.82 |
| SEED-LLaMA | 69.07 | 65.54 |
| SEED-LLaMA-I | 70.68 | 66.55 |
| GILL | 67.45 | 65.16 |
| AnyGPT | 65.00 | - |
| MIO-Base | 64.15 | 62.71 |
| MIO-Instruct | 67.76 | 68.97 |

petitive image generation capabilities. Note that beyond single image generation abilities, our model can also exhibit multi-image generation capabilities such as generating visual stories, image sequences, and even visual thoughts as illustrated in §3.5.

## 3.2 SPEECH-RELATED TASKS

We evaluate the speech understanding and generation abilities of MIO on ASR and TTS tasks. Wav2vec 2.0 (Baevski et al., 2020), Whisper Large V2 (Radford et al., 2023), and AnyGPT (Zhan et al., 2024) are the baselines for ASR tasks, while VALL-E (Wang et al., 2023a), USLM (Zhang et al., 2023b) , and AnyGPT (Zhan et al., 2024) are the baselines for TTS tasks. The test set used for ASR evaluation is LibriSpeech (Panayotov et al., 2015), while the test set used for TTS evaluation is

VCTK (Veaux et al., 2017) following AnyGPT (Zhan et al., 2024)'s practice. The Whisper medium model is used to transcribe the speech generated for the TTS task. The WER (word error rate) is computed by comparing the generated transcribed text with the ground-truth transcription after text normalization[4].

As shown in Table 3.2, our models exhibit speech performance comparable to the speech-specific baselines and outperform the AnyGPT baseline. It is important to note that although AnyGPT is capable of generating content tokens for speech, it lacks the ability to generate timbre tokens, which necessitates the use of an additional voice cloning model. In contrast, our models generate both content and timbre tokens,

Table 5: Speech ability evaluation. "WER" denotes word error rate. Lower values are better.

| Models | ASR WER | Models | TTS WER |
|---|---|---|---|
| Wav2vec | 2.7 | VALL-E | 7.9 |
| Whisper | 2.7 | USLM | 6.5 |
| AnyGPT | 8.5 | AnyGPT | 8.5 |
| MIO-Base | 6.3 | MIO-Base | 12.0 |
| MIO-Instruct | 10.3 | MIO-Instruct | 4.2 |

making the TTS tasks more challenging for our models compared to AnyGPT. Nonetheless, after instruction tuning, our model still achieves better TTS performance. More evaluations of the TTS and Speech-to-Speech generation performance are provided in Appendix E.3 and E.2.

## 3.3 VIDEO-RELATED TASKS

We compare MIO with Flamingo (Alayrac et al., 2022), BLIP-2 (Li et al., 2023b), InstructBLIP (Dai et al., 2023), Emu (Sun et al., 2023c), and SEED-LLaMA (Ge et al., 2023b) for video understanding. The models are evaluated on the MSVDQA (Chen & Dolan, 2011a) and MSRVTT-QA (Xu et al., 2017). The results are presented in Table 6. Our model achieves the highest scores compared to all baselines. Due to the lack of video (frame sequence) generation benchmarks in our setting, we provide video

Table 6: Video understanding evaluation using top-1 accuracy for both benchmarks. "I" denotes the instruction-tuned version.

| Models | MSVDQA | MSRVTT-QA |
|---|---|---|
| Flamingo (9B) | 30.2 | 13.7 |
| BLIP-2 (4.1B) | 33.7 | 16.2 |
| InstructBLIP (8.1B) | 41.8 | 22.1 |
| Emu-Instruct (14B) | 32.4 | 14.0 |
| SEED-LLaMA-I (8B) | 40.9 | 30.8 |
| MIO-Instruct | 42.6 | 35.5 |

generation examples in §3.5. These results demonstrate the superior performance of our models in both video understanding and video generation.

Table 7: Language-only evaluation. "I" denotes the instruction-tuned version.

| Models | MMLU |
|---|---|
| LLAMA-1-7B-Base | 33.0 |
| LLAMA-2-7B-Chat | 47.9 |
| SEED-LLAMA-8B-I | 36.1 |
| AnyGPT-Base | 26.4 |
| AnyGPT-Chat | 27.4 |
| MIO-Instruct | 45.7 |

Table 8: Results for trimodal comprehension (text, image, and speech).

| Models | OmniBench |
|---|---|
| Gemini-1.5-Pro | 42.67 |
| Reka-Core-20240501 | 31.52 |
| AnyGPT (8B) | 17.77 |
| video-SALMONN (13B) | 34.11 |
| Unified-IO 2 (6.8B) | 34.24 |
| MIO-Instruct (7B) | 36.96 |

## 3.4 LANGUAGE-ONLY TASKS

We evaluate our models on MMLU (Hendrycks et al., 2021). The baselines are two LLaMA variants (Touvron et al., 2023a;b), the instruction-tuned SEED-LLaMA (Ge et al., 2023b), and AnyGPT (Zhan et al., 2024). For the MMLU benchmark, we conduct zero-shot evaluation experiments using the official evaluation code. The experimental results are shown in Table 7. We can observe that our models have superior language-only performance compared with all any-to-any MM-LLM baselines and even surpass LLaMA-1-7B-Base, an advanced pure language model.

---

[4] `https://github.com/openai/whisper/blob/main/whisper/normalizers/english.py`

## 3.5 Demonstrations

We illustrate the basic and advanced abilities of MIO in Figure 5 and 4. The basic abilities of MIO involve image understanding and generation, video understanding and generation, ASR, and TTS. The advanced abilities of MIO are based on its any-to-any and multimodal interleaved sequence generation features. These abilities involve visual storytelling (*i.e.*, interleaved video-text generation), chain of visual thought, speech-in-speech-out, instructional image editing, visual guideline generation, etc. We refer the readers to Appendix E.5 for more demonstrations including multimodal chain of thought and multimodal in-context learning.

## 3.6 Ablation Studies

**Generality for Trimodal Understanding.** We evaluate our model using the OmniBench (Li et al., 2024d), which incorporates text, image, and speech modalities as inputs, requiring the model to choose one of four options as the correct answer to determine accuracy. Although MIO acquires its multimodal understanding capabilities through dual-modal training, the evaluation results in Table 8 indicate that MIO also exhibits superior trimodal comprehension abilities.

**Effect of Different Image Tokenizers.** The image tokenizer has a significant impact on image modality alignment. In Figure 2, we compare the image generation performance under a controlled setting after training for solely 3K steps in stage 1, using various image tokenizers. The image tokenizers used for comparison include a VQGAN (Esser et al., 2020) with a vocabulary size of 1024 and a compression rate of 16 (VQGAN-1024), as well as the VQGAN-Gumbel with a vocabulary size of 8192 (VQGAN-8192)[5]. Our results indicate that the SEED-Tokenizer, which captures more semantic and higher-level image information, exhibits faster convergence. In contrast, both VQGAN tokenizers show slower convergence due to their lower-level image information.

## 4 Related Works

### 4.1 Multimodal LLMs

With the rapid success of Large Language Models (LLMs), current multimodal LLMs (MM-LLMs) are typically built on a pre-trained LLM backbone and are endowed with the ability to understand multiple modalities (Li et al., 2019; Lu et al., 2019; Kim et al., 2021; Zeng et al., 2022; Zhou et al., 2022; Wang et al., 2023b; 2024e). Generally, these MM-LLMs align the representations of images obtained from visual encoders with the text embedding space, thereby leveraging the powerful capabilities of the foundational models. For example, BLIP-2 (Li et al., 2023b) uses CLIP-ViT (Radford et al., 2021) to extract high-level features from images and then employs a Q-Former to compress the number of image tokens and further align image tokens

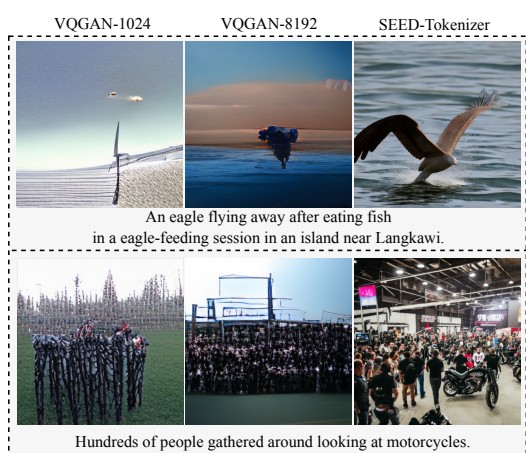

An eagle flying away after eating fish in a eagle-feeding session in an island near Langkawi.

Hundreds of people gathered around looking at motorcycles.

Figure 2: Comparing different image tokenizers for image generation within a controlled setting (limited to 3K training steps).

with the LLM embeddings. In contrast, LLaVA (Liu et al., 2023b; Li et al., 2024a) utilizes a simple linear projection or MLP as the connector between the image encoder and the LLM backbone. These models demonstrate strong multimodal understanding abilities, achieving significant progress in tasks such as visual question answering, visual commonsense reasoning, chart understanding, etc.

Additionally, beyond images, other MM-LLMs have also focused on modalities such as speech and video. For instance, LLaSM (Shu et al., 2023) and InternVideo (Wang et al., 2022; 2024c) are MM-LLMs designed for speech and video understanding, respectively. These models adopt a similar architectural design to BLIP-2 or LLaVA but redesign modality-specific encoders.

---

[5]https://github.com/CompVis/taming-transformers

Recently, increasing attention has been paid to unifying multiple modalities within a single MM-LLM. For example, ImageBind (Girdhar et al., 2023) develops encoders suited for multiple modalities such as images, videos, audio, heat maps, among others, while OmniBind (Wang et al., 2024d) trains an omni-representation model by aligning encoders across four modalities: audio, language, images, and 3D objects. OmniBench (Li et al., 2024d) is proposed to evaluate the models' abilities for visual, acoustic, and textual understanding.

However, these models focus primarily on multimodal understanding and often overlook the important aspect of multimodal generation.

## 4.2 ANY-TO-ANY MM-LLMs

To enable multimodal generation in MM-LLMs, a straightforward approach is to allow these models to call external multimodal generation tools, such as Stable Diffusion (Rombach et al., 2022) or text-to-speech (TTS) tools (Shen et al., 2023; Li et al., 2024c; OpenAI et al., 2023). However, as highlighted in the Gemini technical report (Team et al., 2023), relying on an intermediate natural language interface can limit the model's ability to express images. If a model cannot natively output images, it will not be able to generate images with prompts of interleaved sequences of image and text. This claim is in line with our distinction between descriptive image generation and context-aware image generation, as discussed in §2.2.

As a result, recent works focus on the unification of multimodal understanding and generation in a single model (*i.e.*, any-to-any MM-LLMs), enabling the generation of multimodal tokens without natural language as an interface. These models typically follow different approaches, depending on how images are represented in both input and output sides. For example, the Discrete-In-Discrete-Out (DIDO) approach has been explored in works such as SEED-LLaMA (Ge et al., 2023b), AnyGPT (Zhan et al., 2024), and Chameleon (Team, 2024). Continuous-In-Discrete-Out (CIDO) methods have been implemented in models like DaVinCi (Diao et al., 2023), Gemini (Team et al., 2023), and Unified-IO 2 (Lu et al., 2023). The Continuous-In-Continuous-Out (CICO) approach is used in models such as Emu (Sun et al., 2023c;a), and DreamLLM (Dong et al., 2023). Another approach, the integration of autoregression and diffusion (AR + Diff), can be seen in models like Transfusion (Zhou et al., 2024), Show-o (Xie et al., 2024), and Li et al. (2024b)'s.

However, these models face specific limitations. DreamLLM (CICI, Dong et al. (2023)) and CIDO models suffer from inconsistencies between input and output forms for multimodal data, making it difficult for them to natively support the generation of interleaved multimodal sequences where an image functions in a coupled way as both input and output. Emu2 (CICO, Sun et al. (2023a)) struggles with the challenges of the mean square error (MSE) loss used for training continuous output representations, as well as with the uni-modal assumption of the Gaussian distribution in the MSE loss. Transfusion (AR + Diff, Zhou et al. (2024)) applies noise to images from the input side to support multimodal generation with diffusion modeling, and relies on VAE (Kingma & Welling, 2013) features rather than CLIP (Radford et al., 2021) features for denoising, which largely trade off the multimodal understanding abilities.

To mitigate these issues, we adopt the DIDO approach. A comprehensive comparison of our models with other any-to-any MM-LLMs is presented in Table 1.

## 5 CONCLUSION

In conclusion, MIO represents an advancement in the realm of multimodal foundation models. By employing a rigorous four-stage training process, MIO successfully integrates and aligns discrete tokens across text, image, video, and speech modalities. This comprehensive approach enables MIO to understand and generate multimodal content in an end-to-end, autoregressive manner, addressing the limitations of current multimodal large language models. Our experimental results showcase its competitive performance across a variety of benchmarks compared to the dual-modality baselines and other any-to-any multimodal large language models. With the any-to-any and multimodal interleaved output features, MIO exhibits novel emergent abilities such as interleaved video-text generation, chain-of-visual-thought reasoning, etc.

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

## A  PRE-TRAINING DATA

**Pre-training Data Sources.**    The pre-training data sources involve six types:

1. Image-text paired data: SBU (Ordonez et al., 2011), CC3M (Sharma et al., 2018), LAION-COCO (LAION, 2022), and JourneyDB (Pan et al., 2023), where JourneyDB only serves for image generation.

2. Language-only data: RefinedWeb (Penedo et al., 2023).

3. Image-text interleaved data: OBELICS (Laurençon et al., 2023), MMC4-core-ff (Zhu et al., 2023).

4. Video-text paired data: WebVid-10M (Bain et al., 2021).

5. Video-text interleaved data:  HowTo-100M (Miech et al., 2019), Youtube-Temporal-180M (Zellers et al., 2021).

6. Speech-text paired data: Libriheavy (Kang et al., 2023).

**Pre-training Data Processing.**    We have different data processing procedures for different data types illustrated in §A following Emu (Sun et al., 2023c) and Qwen-VL (Bai et al., 2023):

1. Image-text paired data: we remove pairs with more than 2:1 aspect ratio or smaller than $224 \times 224$ resolution of the image.  We remove pairs with more than 0.27 CLIP scores. We remove non-English pairs. We randomly place the image or text at the forefront for generating captions based on images and vice versa.

2. Language-only data: we use the same data processing pipeline as used in Yi (AI et al., 2024).

3. Image-text interleaved data: we filter the data using a CLIP score threshold of 0.25, and follow the same procedure as illustrated in Emu (Sun et al., 2023c).

4. Video-text paired data: we randomly place the frames or text at the forefront for generating captions based on frames and vice versa. 60% of the pairs are text-to-video, while 40% of the pairs are video-to-text. We sample 4 to 8 frames of each video for training according to the text lengths.

5. Video-text interleaved data: We first use PySceneDetect to extract key frames from the video based on scene changes, following the practice of Stable Video Diffusion (Blattmann et al., 2023). Then, for each video clip between two key frames, we extract a central frame for textual caption generation with BLIP-2 (Li et al., 2023b). Additionally, the video clips between key frames are processed using ASR (automatic speech recognition) tools to extract subtitles. The ASR text and captions are then integrated and refined using Yi-34B-Chat (AI et al., 2024), resulting in a single text segment. These text segments, along with the key frames and central frames, form the video-text interleaved data.

6. Speech-text paired data: we remove speechs with more than 15 seconds.

## B  PRE-TRAINING DETAILS

**Hyperparameters.**    We enable Flash Attention (Dao et al., 2022; Dao, 2023) during pre-training. Gradient clipping is set to 1.0 for all stages. The maximum sequence length for training is 2800 tokens. We use a cosine learning rate scheduler with a peak learning rate of 3e-5 and a warmup ratio of 0.03. The optimizer used is AdamW (Loshchilov & Hutter, 2017).

**Prompt Templates.**    The prompt template is only necessary for paired datasets. For image-text paired data, we use the prompt templates of "{image} The caption of this image is: {caption}" and "Please generate an image of "{caption}": {image}". For video-text paired data: we use the prompt templates of "Please describe the following video: {image} {description}" and "Please generate a video for "{description}": {video}". For speech-text paired data: we use the prompt templates of "{speech} Transcribe this speech: {transcription}" and "Please generate a speech of "{transcription}": {speech}" during Stage I and Stage II. While for Stage III, we change the ASR prompt template into '{speech} The transcription of this speech is: {transcription}".

# C SUPERVISED FINE-TUNING DETAILS

Table 9: Supervised Fine-Tuning Data. "ICL" denotes In-Context Learning, and "CoT" denotes Chain of Thought.

| Task | Dataset |
|---|---|
| **Language Only** | OpenHermes (Teknium, 2023) |
| **Multimodal ICL** | MMICL (Zhao et al., 2023) |
| **Multimodal CoT** | ScienceQA (Lu et al., 2022) |
| **Chart Understanding** | Geo170K (Gao et al., 2023) |
| **Instructional Image Generation** | InstructPix2Pix (Brooks et al., 2023), MagicBrush (Zhang et al., 2024) |
| **ASR** | LibriSpeech (Panayotov et al., 2015), GigaSpeech (Chen et al., 2021), Common Voice (Ardila et al., 2020) |
| **Video Dialogue** | VideoChat2-IT (Li et al., 2023c) |
| **Image QA** | Vision-Flan (Xu et al., 2023), VizWiz (Gurari et al., 2018), LAION-GPT4V[6], LLaVAR (Zhang et al., 2023c), OCR-VQA (Mishra et al., 2019), VQA (Goyal et al., 2016), TextVQA (Singh et al., 2019), OK-VQA (Marino et al., 2019), Mantis-Instruct (Jiang et al., 2024) |
| **Speech Generation** | SpeechInstruct (Zhang et al., 2023a) |
| **Speech Understanding** | SpeechInstruct (Zhang et al., 2023a) |
| **Image Captioning** | Flickr30K (Plummer et al., 2015), MS-COCO (Lin et al., 2014) |
| **Descriptive Image Generation** | Flickr30K (Plummer et al., 2015), MS-COCO (Lin et al., 2014) |
| **TTS** | GigaSpeech (Chen et al., 2021), Common Voice (Ardila et al., 2020) |
| **Video Generation** | MSR-VTT (Xu et al., 2016), MSVD (Chen & Dolan, 2011b) |
| **Video Understanding** | MSR-VTT (Xu et al., 2016), MSVD (Chen & Dolan, 2011b), MSVD-QA (Chen & Dolan, 2011a), MSRVTT-QA (Xu et al., 2017) |
| **Visual Storytelling** | VIST (Huang et al., 2016) |

**Supervised Fine-Tuning Data.** As shown in Table 9, we use 16 tasks with 34 datasets for a comprehensive supervised fine-tuning.

**Prompt Templates.** The chat template is the same as used in Yi (AI et al., 2024). The system prompt is unified as: "You are MIO, an AI assistant capable of understanding and generating images, text, videos, and speech, selecting the appropriate modality according to the context." except for speech generation and TTS whose system prompts are "You are MIO, an AI assistant capable of understanding images, text, videos, and speech, and generating speech. Please respond to the user with speech only, starting with <spch> and ending with </spch>." to avoid randomness of the output modality.

**Hyperparameters.** Similar to pre-training (*c.f.*, Appendix B), we enable Flash Attention (Dao et al., 2022; Dao, 2023) during supervised fine-tuning. Gradient clipping is set to 1.0. The maximum sequence length for training is 2800 tokens. We use a cosine learning rate scheduler with a peak learning rate of 3e-5 and a warmup ratio of 0.03. The optimizer used is AdamW (Loshchilov & Hutter, 2017).

# D EVALUATION DETAILS.

**Hyperparameters.** The decoding strategies and hyperparameters are quite important for a superior performance. As shown in Table 10, we use different sets of parameters for different output modalities.

Table 10: Decoding Hyperparameters.

| Output Modality | Text | Image | Speech | Video |
|---|---|---|---|---|
| Beam size | 5 | 1 | 1 | 1 |
| Do Sampling | False | True | True | True |
| Top-P | - | 0.7 | 0.7 | 0.7 |
| Repetition Penalty | 1.0 | 1.0 | 1.15 | 1.15 |
| Temperature | 1.0 | 1.0 | 1.0 | 1.0 |
| Guidance Scale | 1.0 | 1.0 | 1.0 | 1.0 |

Table 11: Prompt templates used for evaluating instruction-tuned models.

| Task | Prompt Template |
|---|---|
| Image Captioning | Provide a one-sentence caption for the provided image. {image} |
| Image QA | (We use the prompt templates in LMMs-Eval (Li* et al., 2024)). |
| Image Generation | Please generate an image according to the given description. {description} |
| ASR | Please transcribe this speech.{speech_token} |
| TTS | Please generate a speech according to the given transcription. Start with <spch>. {transcription} |
| Text-only | The following are multiple choice questions (with answers) about {subject} {question} |
| Video QA | The goal is to use the visual information available in the image to provide an accurate answer to the question. This requires careful observation, attention to detail, and sometimes a bit of creative thinking.{video} Question: {question} Answer: |

**Prompt Templates.** The prompt templates used for evaluating pre-training checkpoints are the same as used during pre-training. For SFT checkpoint evaluation, we list the prompt templates in Table 11.

# E MORE EXPERIMENTS

## E.1 IMAGE GENERATION EVALUATION

We compute two additional automatic metrics for evaluating image generation, i.e., SSIM (Wang et al., 2004) and Aesthetic Predictor v2.5[7] for the evaluation of structural integrity and aesthetics, respectively. SSIM (Structural Similarity Index Measure) evaluates the perceptual similarity between the generated images and the ground-truth images, focusing on luminance, contrast, and structure, with scores ranging from -1 (dissimilar) to 1 (identical). Aesthetic Predictor V2.5 is a SigLIP (Zhai et al., 2023)-based predictor that evaluates the aesthetics of an image on a scale from 1 to 10 (10 is the best). In addition, we randomly select 100 image descriptions from MS-COCO test set, and used each model to generate images accordingly for human preference evaluation. We ask 3 annotators to rank 3 images generated by the 3 models: "given the image description, which image is preferred?" The average ranking of MIO's, AnyGPT's, and Emu's generated images are 1.2 (MIO), 2.9 (AnyGPT), 1.9 (Emu). MIO aligns the best with the human preference. The percentage agreement between the three annotators (calculated as the number of cases with identical rankings by all annotators divided by 100) is 82.3%, indicating a high consistency in the human evaluation.

| Dataset | MS-COCO | | Flickr30K | | MS-COCO Subset |
| --- | --- | --- | --- | --- | --- |
| Metric | SSIM (↑) | Aesthetic (↑) | SSIM (↑) | Aesthetic (↑) | Human Avg. Ranking (↓) |
| Emu | 0.1749 | 3.733 | 0.1451 | 3.893 | 1.9 |
| AnyGPT | 0.1960 | 3.954 | 0.1585 | 4.251 | 2.9 |
| MIO | 0.2307 | 4.019 | 0.1727 | 4.326 | 1.2 |

Table 12: Image generation evaluation by SSIM, Aesthetic Predictor V2.5, and human preference.

| Model | Supported Workflow | Content Score (1-5 points) (↑) |
| --- | --- | --- |
| MIO | s2s | 1.4 |
| LLaMA-Omni (Fang et al., 2024) | s2t→t2s | 2.4 |
| AnyGPT | s2t→t2s | 1.8 |

Table 13: Speech-to-Speech performance. "s2s" means "speech-to-speech", while "s2t" and "t2s" denote "speech-to-text" and "text-to-speech", respectively.

## E.2 SPEECH-TO-SPEECH EVALUATION

Since there is a lack of speech to speech evaluation benchmarks, we randomly sample some conversations from the moss-002-sft dataset[8] and convert them into speech-to-speech format. Following the evaluation procedures outlined in LLaMA-Omni (Fang et al., 2024), we use the content score metric obtained from GPT-4o (OpenAI et al., 2024) to assess whether the model's response effectively addresses the user's instructions. The results are shown in Table 13.

Though the content score of MIO is slightly lower than LLaMA-Omni and AnyGPT, both LLaMA-Omni and AnyGPT first generate text replies and then convert these into voice. However, our model, MIO, is capable of directly generating speech responses to speech queries.

## E.3 TTS EVALUATION

| Model | GLOBE | | LibriSpeech test-clean | |
| --- | --- | --- | --- | --- |
| | WER (↓) | Speech Similarity (↑) | WER (↓) | Speech Similarity (↑) |
| MIO | 9.8 | 67.8 | 10.3 | 75.1 |
| AnyGPT | 27.9 | 67.3 | 28.1 | 71.3 |

Table 14: More automatic evaluations for the TTS performance.

We select two additional benchmarks, LibriSpeech test-clean (Panayotov et al., 2015) and GLOBE (Wang et al., 2024b), to evaluate the performance of TTS between our model and AnyGPT. For fair comparison, we don't specify the input voice prompt during evaluation of MIO and AnyGPT. WER (Word Error Rate) and speaker similarity are employed as the automatic metrics. The results are shown in Table 14. The results show that MIO performs significantly better than AnyGPT on both WER and speaker similarity across both benchmarks.

Additionally, we conduct a human evaluation to assess the speech quality of the outputs from MIO and AnyGPT. In this evaluation, participants are provided with the target speech, the speech generated by AnyGPT, and the speech generated by our model. They are tasked with determining which one sounded more natural and closer to the target speech. Evaluators could choose one of the two generated speeches or indicate that they find them equally natural.

Table 15: Human evaluation for the TTS performance.

| | |
| --- | --- |
| MIO Win | 54% |
| Tie | 25% |
| MIO Lose | 21% |

---

[7]https://github.com/discus0434/aesthetic-predictor-v2-5?tab=readme-ov-file

[8]https://huggingface.co/datasets/fnlp/moss-002-sft-data

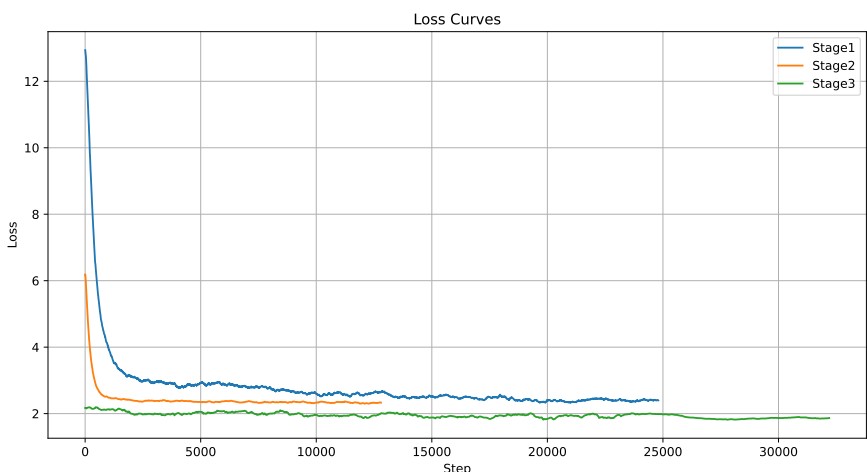

Figure 3: Loss curves of pretraing stages.

Each evaluation is rated by three independent human evaluators, and we report the average scores. The results are shown in Table 15. MIO significantly outperforms AnyGPT in the human evaluation, consistent with the results from the automatic evaluation.

### E.4 LOSS CURVES

We plot the loss curves for each stage in Figure 3. We can observe that when introducing a new data type (i.e., image-text interleaved data) in stage 2, the training loss suddenly increases. However, in the third pretraining stage, i.e., the speech-enhancement stage, the training loss transitions more smoothly. Despite the fluctuations in loss between stages, which do have some impact on downstream performance during the fluctuation periods, we find that with continued training, the model's loss quickly recovers to its previous convergence level and continues optimizing effectively.

### E.5 MORE DEMONSTRATIONS.

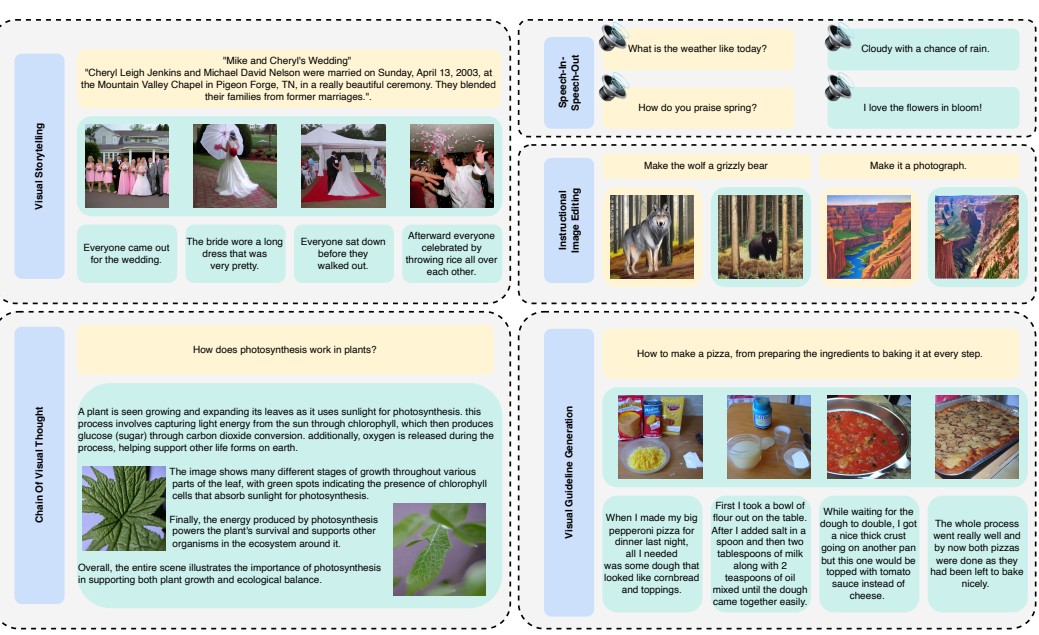

Figure 4: Demonstrations of MIO's advanced abilities. Yellow : inputs; Green : outputs.

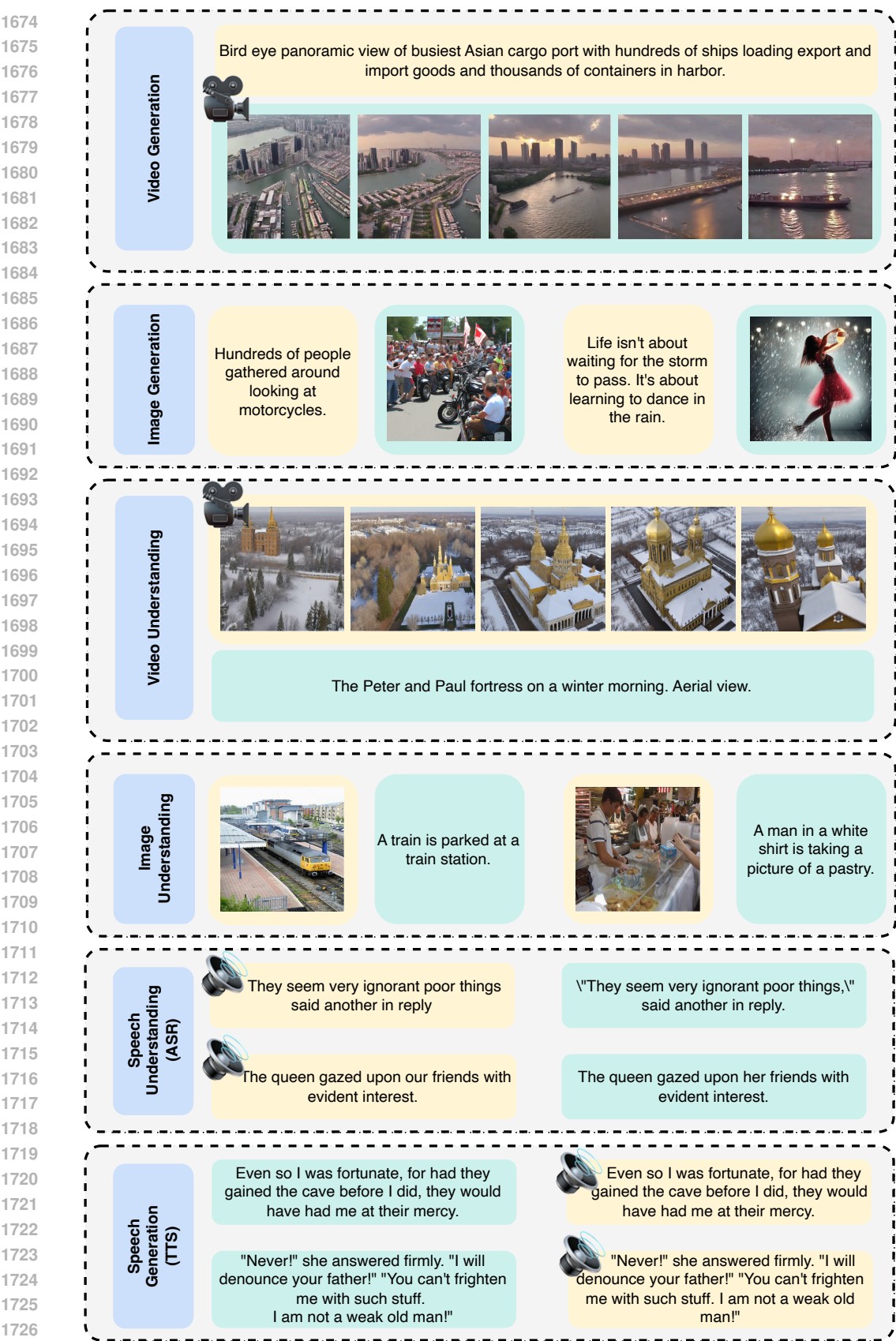

Figure 5: Demonstrations of MIO's basic abilities. Yellow: inputs; Green: outputs.

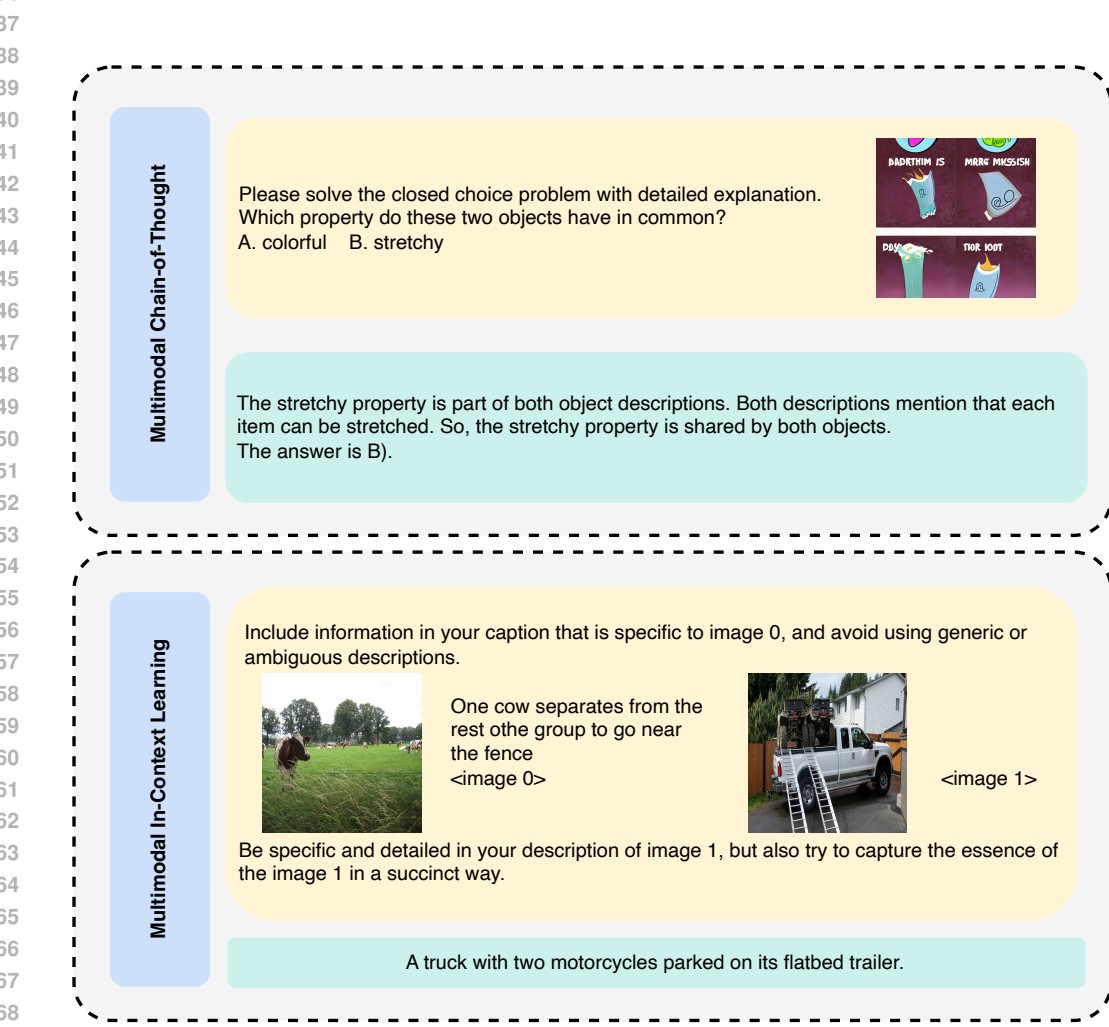

Figure 6: Multimodal Chain-of-Thought and Multimodal In-Context Learning Demos. Yellow: inputs; Green: outputs.

