# OpenReview forum: "MIO: A Foundation Model on Multimodal Tokens"
_ICLR.cc/2025/Conference — Submitted to ICLR 2025_

### Official Review · Reviewer_GVGf · 2024-11-02

**Soundness:** 3
**Presentation:** 2
**Contribution:** 3
**Rating:** 6
**Confidence:** 4

**Summary:**

The paper introduces MIO, a novel open-source, any-to-any multimodal foundation model designed for understanding and generating content across four modalities: text, images, speech, and video keyframes. Built with a Discrete-In-Discrete-Out (DIDO) approach, MIO leverages pretrained multimodal tokenization and a four-stage training pipeline to ensure robust cross-modality performance.

**Strengths:**

1. MIO introduces a novel approach as a foundation model tailored for interleaved cross-modality generation, advancing current capabilities in multimodal large language models.

2. By utilizing pretrained multimodal tokenizers, MIO enables scalable and efficient parallel training on interleaved multimodal data, enhancing its adaptability across diverse input types.

3. As a foundation model, MIO offers substantial potential for further research and development in multimodal large language models and related downstream tasks.

**Weaknesses:**

1. Qualitative comparisons with other baselines are missing. One of the main differences between MIO and Emu is the change from CICO to DIDO, and Table 4 shows that MIO performs better in image generation. The paper can be enhanced by showing the qualitative comparisons with other models, like Emu.

2. Need more clarification on multimodal inference. During the inference time, how does the model output fix-length image tokens (like one image should have 32 tokens) and speech tokens, especially for videos? Furthermore, do authors use any strategies to promise the multimodal tokens within their vocabulary?

3. Need more explanation on results. From table 3, MIO has better performance on image understanding with fewer parameters (7B vs 80B) even compared with some works that mainly focus on image understanding, like InstructBLIP and Flamingo. Can authors analyze which part led to this outperformance? Also, MIO outperforms Emu on several benchmarks. Does this indicate DIDO is better than CICO in both image generation and understanding?

4. The video generation in MIO seems more like video keyframe generation, which only maintains semantic and style consistency.

**Questions:**

1. One related work, NextGPT [1], is missing, which also can do text, speech, and video generation. Can authors please claim the difference between MIO and this work?

2. Why is Emu2 missing in the results?

[1] Wu, Shengqiong, et al. "Next-gpt: Any-to-any multimodal llm." arXiv preprint arXiv:2309.05519 (2023).

---

> ### Author Response · Authors · 2024-11-30
> **Responses for Q1 - Q5**
>
> Thanks for your kind reviews. We provide our clarifications below:
>
> Q1: qualitative comparisons between MIO and Emu for image generation.
>
> A1: We have provided the qualitative comparison in the supplemental materials: `MIO supplemental materials\IMAGES-qualitative comparisions bwteen MIO and Emu for image generation.zip`. The comparisons demonstrate the superior image generation abilities of our model.
>
> ---
>
> Q2: how does the model output fix-length image tokens, speech tokens, and video tokens? do authors use any strategies to promise the multimodal tokens within their vocabulary?
>
> A2: During training, we use special tokens such as `<image>`, `</image>`, `<spch>`, and `</spch>` to define the boundaries for each modality data (images, video frames, and speeches). Specifically:
>
> - **`<image>` and `</image>`**: These tokens mark the beginning and end of an image or a video frame.
> - **`<spch>` and `</spch>`**: These tokens are used to define the start and end of a speech.
>
> During training, these tokens help the model distinguish between different modalities and learn the patterns, such as how many tokens each image or speech should be represented by. For example, the model may learn that an image is typically represented by 32 tokens, and that the number of speech tokens for each codebook should be consistent. These patterns regarding the token length can be automatically learned by the models.
>
> Moreover, these multimodal tokens are added directly to the model's vocabulary. The embedding matrix and language modeling head are extended to accommodate the increased vocabulary size, supporting these new multimodal tokens.
>
> So during inference, we don’t use any special trick to promise the multimodal tokens. No constraint is leveraged for generation.
>
> ---
>
> Q3: The reason for better image understanding? Is DIDO better than CICO considering that MIO > Emu in many benchmarks?
>
> A3: The better performance of MIO on image understanding with fewer parameters can be attributed to more semantically aligned image tokens and cleaner, richer training data.
>
> Although MIO outperforms Emu in several benchmarks, we cannot conclusively claim that DIDO is better than CICO in all scenarios. The VQ-based tokenization in DIDO may incur some information loss, which could affect performance in tasks where fine-grained details matter. It's also worth noting that the latest version of Emu3, which adopts DIDO instead of CICO, has achieved strong results, indicating DIDO's strong potential. However, in image understanding tasks, CI-based approaches still tend to perform better, while in image generation, CO might struggles with the challenges of the mean square error (MSE) loss used for training continuous output representations, as well as with the uni-modal assumption of the Gaussian distribution in the MSE loss, as discussed in our related works.
>
> [1] Emu3: Next-Token Prediction is All You Need
>
> ---
>
> Q4: only video keyframe generation, not a real video.
>
> A4: Video keyframe generation is also a vital issue because the keyframes can capture the most important visual moments of a video, serving as a skeleton for further full video generation with richer semantics. Even the most advanced full video generation models face difficulties in planning and generating keyframes, which are the critical points that define the overall structure and narrative of a (long) video. Compared with full video generation, which focuses on the temporal dynamics and frame density, video keyframe generation focuses on generating the representative frames to establish coherent and meaningful transitions between key moments in the video. These keyframes provide a high-level outline of the video’s content, allowing for easier synthesis of more complex narratives while maintaining semantic consistency across frames. We emphasize this capability over full video generation, as it allows for more effective control and planning of video content.
>
> ---
>
> Q5: the difference between MIO and NeXT-GPT
>
> A5: The differences between NeXT-GPT and MIO can be summarized as follows:
>
> 1. **Approach**: NeXT-GPT uses a CICO (continuous-in, continuous-out) method, while MIO adopts a more flexible DIDO (discrete-in, discrete-out) approach.
> 2. **Consistency of I/O**: In NeXT-GPT, the input and output representations for the same image/audio data are not consistent, whereas MIO ensures consistency across these representations.
> 3. **Multimodal Interleaving**: NeXT-GPT does not support multimodal interleaved output, whereas MIO can handle the interleaved generation of multiple modalities more effectively due to the I/O consistency feature and the multimodal interleaved training stage.

---

> ### Author Response · Authors · 2024-11-30
> **Responses for Q6**
>
> Q6: Why is Emu2 missing in the results?
>
> A6: The Emu2 results have been added in the following tables:
>
> - image understanding:
>
> | **Models** | Imagen | Speech | COCO | VQAv2 | OKVQA | VizWiz | SEED Bench |
> | --- | --- | --- | --- | --- | --- | --- | --- |
> | Emu-Base (14B) | ✔️ | ❌ | 112.4 | 52.0 | 38.2 | 34.2 | 47.3 |
> | Emu-I (14B) | ❌ | ❌ | 120.4 | 57.2 | 43.4 | 32.2 | 58.0 |
> | SEED-LLaMA-I (8B) | ✔️ | ❌ | 124.5 | 66.2 | 45.9 | 55.1 | 51.5 |
> | AnyGPT (8B) | ✔️ | ✔️ | 107.5 | - | - | - | - |
> | Flamingo (9B) | ❌ | ❌ | 79.4 | 51.8 | 44.7 | 28.8 | 42.7 |
> | Flamingo (80B) | ❌ | ❌ | 84.3 | 56.3 | 31.6 |  | - |
> | Kosmos-1 (1.6B) | ❌ | ❌ | 84.7 | 51.0 | - | 29.2 | - |
> | MetaLM (1.7B) | ❌ | ❌ | 82.2 | 41.1 | 11.4 | - | - |
> | IDEFICS-I (80B) | ❌ | ❌ | 117.2 | 37.4 | 36.9 | 26.2 | 53.2 |
> | CM3Leon (7B) | ✔️ | ❌ | 61.6 | 47.6 | 23.8 | 37.6 | - |
> | InstructBLIP (8.1B) | ❌ | ❌ | - | - | - | 34.5 | 58.8 |
> | Emu2-Base (37B)* | ✔️ | ❌ | - | 68.8 | 57.1 | 57.0 | - |
> | Emu2-Chat (37B)* | ❌ | ❌ | - | 84.9 | 64.8 | 54.9 | 62.8 |
> | **MIO-Instruct (7B)** | ✔️ | ✔️ | 120.4 | 65.5 | 39.9 | 53.5 | 54.4 |
>
> **Note that the reported results of Emu2-Base are evaluated in 16-shot setting.*
>
> - image generation
>
> | **Models** | MS-COCO | Flickr30K |
> | --- | --- | --- |
> | Emu-Base | 66.46 | 64.82 |
> | SEED-LLaMA | 69.07 | 65.54 |
> | SEED-LLaMA-I | 70.68 | 66.55 |
> | GILL | 67.45 | 65.16 |
> | AnyGPT | 65.00 | - |
> | Emu2-Gen | 68.6 | - |
> | MIO-Base | 64.15 | 62.71 |
> | MIO-Instruct | 67.76 | 68.97 |
> - video understanding
>
> | **Models** | MSVDQA | MSRVTT-QA |
> | --- | --- | --- |
> | Flamingo (9B) | 30.2 | 13.7 |
> | BLIP-2 (4.1B) | 33.7 | 16.2 |
> | InstructBLIP (8.1B) | 41.8 | 22.1 |
> | Emu-Instruct (14B) | 32.4 | 14.0 |
> | SEED-LLaMA-I (8B) | 40.9 | 30.8 |
> | Emu2-Chat | 49.0 | 31.4 |
> | MIO-Instruct | 42.6 | 35.5 |
>
> We find that although MIO-Instruct underperforms Emu2 for image understanding, it achieves competitive results in video understanding and image generation with a significantly smaller model size. Moreover, the instruction-tuned variants of Emu2, namely, Emu2-Chat and Emu2-Gen, separate their abilities for image understanding and image generation. That is, Emu2-Chat doesn’t support image generation, and Emu2-Gen doesn’t support image understanding. While our MIO-Instruct integrates the two directions in one single model, even with the support for speech modality which is heterogeneous to the vision modalities.
>
> ---
>
> Thanks again for your valuable feedback and suggestions. We are looking forward to your further feedback.

---

> ### Author Response · Authors · 2024-12-02
> **Looking forward to further feedback**
>
> Hi, Reviewer GVGf,
>
> Thank you for sharing your valuable feedback. As the discussion period approaches its conclusion, we would like to know if our responses have effectively addressed your concerns.

---

### Official Review · Reviewer_Uas7 · 2024-11-02

**Soundness:** 2
**Presentation:** 3
**Contribution:** 2
**Rating:** 5
**Confidence:** 4

**Summary:**

This paper proposes a novel multimodal foundation model named "MIO" to understand and generate speech, text, images, and videos in an end-to-end, autoregressive manner. MIO is built on multimodal tokens across four modalities and undergoes a four-stage training process, including alignment pre-training, interleaved pre-training, speech-enhanced pre-training, and comprehensive supervised fine-tuning on diverse textual, visual, and speech tasks. The model demonstrates competitive performance compared to previous models and showcases advanced capabilities such as interleaved video-text generation and chain-of-visual-thought reasoning.

**Strengths:**

1. Similar to GPT-4o, the proposed "MIO" model is capable of understanding and generating multimodal contents across text, image, speech, and video modalities, which is a notable advancement.

2. To address the disparity of different modalities, the paper proposes a three-stage pretraining process, including alignment, interleaving, and speech ability enhancement. Experiments show that with such a design, the model can generate various modality contents in a unified model.

3. MIO exhibits advanced features such as interleaved video-text generation and visual guideline generation.

4. The authors will release the model, which has the potential to contribute to society.

**Weaknesses:**

1. This paper heavily relied on each modality's existing tokenizer or detokenizer, so it is restricted by their drawbacks. For example, SpeechTokenizer, although it provides a good semantic and acoustic discrete representation of a waveform, is RVQ-based and has many codebooks. So speech generation is inefficient (with 200hz).

2. As the paper claims multi-modal tokens for LLM, although it focuses on video/speech/image understanding and generation, it is necessary to evaluate the text-related benchmarks, i.e., MMLU, MMLU-Pro, GPQA etc. Please follow the practice of Llama[1] and Qwen[2] and compare your models with similar open-source models (e.g., Llama3 8B).
- [1] Dubey, Abhimanyu, et al. "The llama 3 herd of models." arXiv preprint arXiv:2407.21783 (2024).
- [2] Yang, An, et al. "Qwen2 technical report." arXiv preprint arXiv:2407.10671 (2024).

3. The authors propose a three-stage pretraining mechanism with a low ratio to a high ratio of speech tokens to tackle the disparity among different modalities.
* 3.1 Please conduct ablation experiments to demonstrate the effectiveness of the proposed method. In detail, you can compare the three-stage pretraining with one-stage pretraining. Then the one-stage pretraining's speech ratio can vary from low to high. If pretraining on the whole dataset is unaffordable, you can conduct it on a reasonable subset.
* 3.2 Usually changing the training data ratio may cause unstable results. For example, from stage 1 to stage 3, the text ratios decrease largely, thus leading to the degradation of text generation ability. Can you provide the detailed training/validation loss curve of each modality for each stage? Some evaluation results such as teacher-forcing ppl (or you can follow the practice of llama3[1] to evaluate the performance of pretrained model) are also helpful.
- [1] Dubey, Abhimanyu, et al. "The llama 3 herd of models." arXiv preprint arXiv:2407.21783 (2024).

4. For speech, the paper only conducts experiments on TTS/ASR. But as the paper claims to address the gap of GPT-4o, the most interesting and important ability, speech dialogue, is not included. In TTS/ASR tasks, the model is given textual/speech content mapping, so they can not demonstrate speech ability like GPT-4o. So I think it is an over-claim.

Furthermore, you can conduct speech-to-speech task (an example is llama-omini[1]), and evaluate it on speech-to-speech benchmarks [1]
- [1] Fang, Qingkai, et al. "Llama-omni: Seamless speech interaction with large language models." arXiv preprint arXiv:2409.06666 (2024).

5. For ASR experiments, the results are not good enough. WER 6.3 is relatively high[1]. I think the first layer of SpeechTokenizer may lose some semantic information. Why did the instructed model perform worse (6.3 --> 10.3)?

- [1] https://github.com/espnet/espnet/blob/master/egs/librispeech/asr1/RESULTS.md

6. For TTS experiments, the experiments are not convincing. I recommend following the evaluation practice of zero-shot TTS experiments[1,2,3], including 1) speaker similarity, 2) speech quality, 3) robustness (WER), and some other aspects. I also recommend evaluating on more than one benchmark.

- [1] Chen, Sanyuan, et al. "VALL-E 2: Neural Codec Language Models are Human Parity Zero-Shot Text to Speech Synthesizers." arXiv preprint arXiv:2406.05370 (2024).
- [2] Le, Matthew, et al. "Voicebox: Text-guided multilingual universal speech generation at scale." Advances in neural information processing systems 36 (2024).
- [3] Ju, Zeqian, et al. "Naturalspeech 3: Zero-shot speech synthesis with factorized codec and diffusion models." arXiv preprint arXiv:2403.03100 (2024).
- [4] Jiang, Ziyue, et al. "Mega-TTS 2: Boosting Prompting Mechanisms for Zero-Shot Speech Synthesis." The Twelfth International Conference on Learning Representations. 2024.

7. It is difficult to understand the demonstration cases, especially for speech (i.e., I can not listen to the real samples). I recommend preparing an anonymous demo page instead of only several pictures for the case study. An example is https://aka.ms/valle2 for VALL-E 2.

**Questions:**

Please see the weakness.

---

> ### Author Response · Authors · 2024-11-30
> **Responses for Q1 - Q2**
>
> Q1: limited by the tokenizers (esp., SpeechTokenizer, which is inefficient).
>
> A1: We appreciate your insightful feedback regarding the tokenization approach in MIO, especially concerning the inefficiencies of the SpeechTokenizer.
>
> First, we would like to highlight that our approach integrates **voice** (acoustic signals) into the language model, not just the content (semantic signals). By training not only on the semantic aspects of speech but also on its acoustic properties, our model gets rid of the additional needs of a voice cloning model compared with the previous works such as AnyGPT.
>
> Regarding the inefficiency of the SpeechTokenizer, we acknowledge that the current RVQ-based tokenization with multiple codebooks leads to suboptimal performance, particularly in terms of generation speed (200 Hz). However, more efficient and effective speech tokenizers than SpeechTokenizer, such as WavTokenizer[1] and GLM-4-Voice-Tokenizer[2] are our concurrent works. We plan to replace our speech tokenizer by these more efficient ones in the next version of our model.
>
> [1] WavTokenizer: An Efficient Acoustic Discrete Codec Tokenizer for Audio Language Modeling
>
> [2] GLM-4-Voice-Tokenizer: https://github.com/THUDM/GLM-4-Voice
>
> ---
>
> Q2: text-only benchmarks.
>
> A2: We evaluate our models on MMLU, a well-known language-only benchmark. The results are shown below:
>
> | Models | MMLU |
> | --- | --- |
> | LLAMA-1-7B-Base | 33.0 |
> | LLAMA-2-7B-Base | 45.3 |
> | LLAMA-2-7B-Chat | 47.9 |
> | SEED-LLAMA-8B-I | 36.1 |
> | AnyGPT-Base | 26.4 |
> | AnyGPT-Chat | 27.4 |
> | MIO-7B-Base | 39.1 |
> | MIO-7B-Instruct | 45.7 |
>
> We can observe that our models have superior language-only performance compared with all any-to-any MM-LLM baselines and even surpass LLaMA-1-7B-Base, an advanced pure language model.

---

> ### Author Response · Authors · 2024-11-30
> **Responses for Q3.1**
>
> Q3.1: three-stage pretraining v.s. one-stage pretraining with speech ratio warmup (on a subset)
>
> A3.1: Thanks for your insightful suggestions!! This is a good idea. The proposed one-stage pretraining (i.e., continuous speech ratio warmup) is conceptually similar to our three-stage strategy. Both aim to mitigate the imbalance caused by the extremely large number of speech tokens compared to other modalities by increasing the speech data ratio step by step. The core difference is, our three-stage strategy implements a **staircase-style warmup**, where the speech data ratio increases suddenly after numerous training steps. In contrast, the proposed one-stage pretraining method uses a **continuous ratio warmup**, where the speech data ratio increases gradually after each training step.
>
> To validate the effectiveness of the three-stage approach, we compare stage 1, stage 2, and stage 3, with each stage being independent and from scratch with the corresponding data mixing ratio (note that “stage 2” is different from “stage 1+2”). After training for 1K to 2K steps, the experimental results are shown below:
>
> | Setup | Data Composition | COCO Captioning (CIDEr) | COCO Imagen (human evaluation) | LibriSpeech ASR (WER) |
> | --- | --- | --- | --- | --- |
> | stage 1 (1K) | 12:2:0:2 | 10.84 | vaguely relevant to the caption and with a clear outline of the described object. | 90.19 |
> | stage 2 (1K) | 2:2:6:6 | 4.21 | low-quality images, irrelevant to the caption. | 89.90 |
> | stage 3 (1K) | 2:1:1:12 | 2.80 | low-quality images, irrelevant to the caption. | 85.32 |
> | stage 1 (2K) | 12:2:0:2 | 25.03 | vaguely relevant to the caption and with a clear outline of the described object. | 90.03 |
> | stage 2 (2K) | 2:2:6:6 | 4.53 | low-quality images, irrelevant to the caption. | 88.02 |
> | stage 3 (2K) | 2:1:1:12 | 2.86 | low-quality images, irrelevant to the caption. | 80.73 |
>
> Note that the data composition is in format of “(image-text pair data) : (language-only data) : (image-text interleaved data + video data) : (speech-text pair data)”
>
> We can observe that the image-text alignment by “stage 2”-only improves much slower than by “stage 1”. And there is a risk for very early performance saturation for image-text alignment when directly using the data mixture of stage 2. Moreover, since the image-text paired corpus is much larger and higher-quality than the image-text interleaved corpus, to ensure the pretraining can consume most of the image-text paired data, we exclude the image-text interleaved data in the first stage. And to further unlock the models’ abilities for image-text interleaved generation, we add these data back into the pretraining once the training loss of the stage 1 nearly converges. As for the third stage — speech enhancement pretraining, since the speech tokens are much more enormous, we maximize its data ratio to ensure most of the speech data is consumed once the performance for other modalities during stage 2 is saturated.
>
> In conclusion, such staircase-style warmup is designed to avoid early performance saturation for image-text representation alignment and to ensure that most of the training data for each modality is trained. And this staged strategy is effective, which can further demonstrate that the speech ratio warmup strategy is effective in a staircase-style setting.
>
> We are very interested in the idea of “speech ratio warmup in a continuous setting” and are eager to validate this approach in a future version of our work. We believe this represents a novel idea in pretraining and could potentially merit an entirely new paper. However, in the current version, resource and time limitations prevent us from conducting a fair and comprehensive comparison between the staircase-style and continuous warmup strategies. Even when using a subset of the data, such experiments would require approximately one week of pretraining **per setup** to produce reliable results. Conducting **both** strategies **simultaneously** for a fair comparison would further increase resource demands. Moreover, shorter pretraining durations would likely fail to yield meaningful results, as transitioning to the later stage before the early-stage loss has sufficiently converged would significantly increase the risk of training instability and unreliable findings. We look forward to exploring this promising idea in our future work.

---

> ### Author Response · Authors · 2024-11-30
> **Responses for Q3.2 - Q4**
>
> Q3.2: loss curves of each modality for each stage? teacher-forcing ppl or other evaluation (e.g., LLaMA3’s) of the pretrained model? Especially for text-only performance.
>
> A3.2: We find that the losses of different modalities decrease alongside the total loss, so we haven’t logged the losses for each modality to reduce the cost for calculating validation losses. However, we can still provide the training losses for each stage. The loss curves are shown in Figure 3. We observe that when a new data type (i.e., image-text interleaved data) is introduced in stage 2, the training loss exhibits a sudden increase. In contrast, the third pretraining stage, which does not introduce novel data types (i.e., the speech-enhancement stage), demonstrates a smoother transition in training loss.
>
> Despite the fluctuations in loss between stages, which do have some impact on downstream performance during the fluctuation periods, we find that with continued training, the model's loss quickly recovers to its previous convergence level and continues optimizing effectively. As noted in A3.1, this staged strategy is both necessary and effective, making such fluctuations negligible in practice. Additionally, we agree that the proposed one-stage pretraining method with continuous speech ratio warmup could potentially avoid these fluctuations and result in more stable training. However, as mentioned in A3.1, we look forward to exploring this promising idea in our future work, once resource limitations and the need for rigorous experimentation can be addressed.
>
> ---
>
> Q4: evaluating speech dialogue performance.
>
> A4: Since the speech-to-speech benchmark of LLaMA-Omni is not open-sourced, we randomly sample some conversations from the moss-002-sft dataset and convert them into speech-to-speech format. Following the evaluation procedures outlined for LLaMA-Omni, we use the content score metric obtained from GPT-4o to assess whether the model's response effectively addresses the user's instructions. The results are as follows:
>
> | Model | Supported Workflow | Content Score(1-5 points) |
> | --- | --- | --- |
> | MIO | s2s | 1.4 |
> | LLaMA-Omni | s2t→t2s | 2.4 |
> | AnyGPT | s2t→t2s | 1.8 |
>
> Note that “s2s” means “speech-to-speech”, while “s2t” and “t2s” denote “speech-to-text” and “text-to-speech”, respectively. Though the content score of MIO is slightly lower than LLaMA-Omni and AnyGPT, both LLaMA-Omni and AnyGPT first generate text replies and then convert these into voice. However, our model, MIO, is capable of directly generating speech responses to speech queries.
>
> **note that LLaMA-Omni is also a submission for ICLR 2025 (id=Submission2959), but we are happy to compare with it.*

---

> ### Author Response · Authors · 2024-11-30
> **Responses for Q5 - Q6**
>
> Q5: limited ASR performance, and worse for instruction-tuned variant.
>
> A5:
>
> 1. The ASR performance of MIO, as shown in Table 5, is competitive for a general-purpose multimodal foundation model. While models like Whisper and Wav2Vec achieve lower WER values due to their specialization in ASR tasks, MIO-Base achieves a WER of 6.3, outperforming AnyGPT’s 8.5. Both MIO and AnyGPT are designed for multiple modalities, including text, images, speech, and video, but MIO demonstrates superior ASR performance, highlighting its balanced optimization across modalities while maintaining strong speech-to-text capabilities.
> 2. The SpeechTokenizer employs multiple layers of residual vector quantization (RVQ) to discretize speech, with the first layer specifically incorporating a HuBERT-based distillation to enhance the representation of content and semantic information. Subsequent layers are optimized using a reconstruction loss to extract timbre signals, effectively dividing the tokenization process into a content codebook (first layer) and timbre codebooks (subsequent layers). However, despite this design for content-timbre disentangling, the first layer still suffers from some semantic loss. This is partly due to HuBERT's inherent limitations in capturing the full range of speech semantics and partly because the SpeechTokenizer does not achieve complete content-timbre disentanglement since the later RVQ layers, which are primarily focused on timbre, still retain traces of semantic information, leading to a slight degradation in the content representation provided by the first layer.
> 3. The instruction-tuned variant of MIO performs worse on ASR (WER 10.3 vs. 6.3 for MIO-Base) primarily due to the shift in task distribution during supervised fine-tuning. In the pretraining stage, ASR-related tasks constitute a significant proportion of the training data, allowing the model to develop robust speech-to-text capabilities. However, during SFT, the inclusion of a broader range of multimodal tasks dilutes the ASR-specific focus. Furthermore, we observed that the instruction-tuned variant tends to extend transcription outputs or generate additional content beyond the transcription instruction. This behavior likely arises from the model's exposure to tasks involving contextual continuation, which inadvertently impacts its ability to adhere strictly to ASR instructions. However, the ASR performance of the instruction-tuned model is still comparable to that of AnyGPT.
>
> ---
>
> Q6: more metrics and more benchmarks for TTS:
>
> A6: We have selected two additional benchmarks, LibriSpeech test-clean[1] and GLOBE[2], to evaluate the TTS performance between our model and AnyGPT. For fair comparison, we did not specify the input voice prompt during the evaluation of MIO and AnyGPT. WER and speaker similarity are used as the automatic metrics. The results are shown below:
>
> | Model | GLOBE (WER) | GLOBE (Speech similarity) | LibriSpeech test-clean (WER) | LibriSpeech test-clean (Speech similarity) | VCTK (WER) | VCTK (Speech similarity) |
> | --- | --- | --- | --- | --- | --- | --- |
> | MIO-Base | 30.3 | 69.1 | 18.2 | 75.4 | 12.0 | 65.1 |
> | AnyGPT-Base | 27.9 | 67.3 | 28.1 | 71.3 | 8.5 | 61.1 |
> | MIO-Instruct | 9.8 | 67.8 | 10.3 | 75.1 | 4.2 | 62.2 |
>
> The results show that MIO performs significantly better than AnyGPT on both WER and speaker similarity across all benchmarks. (Note that AnyGPT-Instruct doesn’t support TTS.)
>
> Additionally, we conduct a human evaluation to assess the speech quality of the outputs from MIO and AnyGPT. In this evaluation, we randomly select 100 samples. Participants are provided with the target speech, the speech generated by AnyGPT, and the speech generated by our model. They are tasked with determining which one sounded more natural and closer to the target speech. Evaluators could choose one of the two generated speeches or indicate that they find them equally natural. Each evaluation is rated by three independent human evaluators, and we report the average scores.
>
> | MIO Win | Tie | MIO Lose |
> | --- | --- | --- |
> | 54% | 25% | 21% |
>
> MIO significantly outperforms AnyGPT in the human evaluation, consistent with the results from the automatic evaluation.
>
> [1] Panayotov, Vassil, et al.  Librispeech: an asr corpus based on public domain audio books. *ICASSP* 2015.
>
> [2] Wang, Wenbin, Yang Song, and Sanjay Jha. GLOBE: A High-quality English Corpus with Global Accents for Zero-shot Speaker Adaptive Text-to-Speech. Interspeech 2024.

---

> ### Author Response · Authors · 2024-11-30
> **Responses for Q7**
>
> Q7: difficult demos, and cannot hear the audio.
>
> A7: For the demonstrations in Figures 4, 5, and 6, there are three elements for each example: instruction (blue, vertical), input (yellow, horizontal), and output (green, horizontal). We replace the detailed instructions with the task names for simplicity. For example, for the bottom-right example in Figure 4, the task is “visual guideline generation”. The model was asked to generate a visual guideline for “how to make a pizza, from preparing the ingredients to baking it at every step.”. And the model generated the images and texts marked with light green background.
>
> As for the speech demonstrations, we have uploaded the audio files in the supplemental materials. See `MIO supplemental materials\MIO_anonymous\generated_speeches`. You can also refer to `MIO supplemental materials\MIO_anonymous\infer.py` for the detailed inference examples.
>
> ---
>
> Thanks again for your valuable feedback and interesting ideas. Your comments are the most valuable ones that I have ever seen. Hope our responses can address most of your concerns.

---

> > ### Comment · Reviewer_Uas7 · 2024-12-03
> >
> > Thanks for your detailed reply and they addressed most of my concerns! This is a valuable and interesting attempt to develop a multi-modal foundation model.  I appreciate the authors' efforts.
> >
> > However, 1. I have listened to the TTS samples and found them in low quality (e.g., detokenized_speech_2_0.wav in instruct_tts is broken, and the speech is too fast). I think the speech generation should be promoted. 2. For speech2speech, how do you conduct speech continuation for speech-to-speech generation? Because you only include Libriheavy which is an ASR corpus and does not contain speech dialogue data.
> >
> > Considering the insufficient results in speech-related tasks, I decide to keep the original score.

---

> > > ### Author Response · Authors · 2024-12-03
> > > **Explanations regarding the speech-related performance**
> > >
> > > Thanks for your further feedback.
> > >
> > > 1. The speech demonstrations are not cherry-picked. We acknowledge that there may be cases where the speech is broken or excessively fast. However, in the majority of cases, the generated speech remains understandable and normal. Concerning speech speed, although our generated speech tends to be faster than the average human speaking speed, it still falls within a range that is both comprehensible to humans and feasible for them to articulate. Additionally, it is important to note that our model is not specifically or exclusively designed for speech-related tasks. Our model can support multiple modalities simultaneously within a single model.
> > >
> > > 2. Libriheavy is a paired speech-transcription dataset. We construct both speech-to-text and text-to-speech data from this paired dataset to support speech generation during pretraining. Furthermore, we integrate additional speech-to-speech data during the instruction tuning phase, such as SpeechInstruct, which includes speech-to-speech generation data.

---

> ### Author Response · Authors · 2024-12-02
> **Looking forward to further feedback**
>
> Hi, Reviewer Uas7,
>
> We sincerely appreciate your valuable feedback. As the discussion period draws to a close, we kindly ask if you could confirm whether our responses have adequately resolved your concerns.

---

### Official Review · Reviewer_dfgg · 2024-11-03

**Soundness:** 3
**Presentation:** 3
**Contribution:** 3
**Rating:** 6
**Confidence:** 4

**Summary:**

This paper introduces a new multimodal foundation model called MIO. Multimodal foundation models are essential tools for related research, similar to the role of SD in text-to-image tasks. Although this resembles a systems engineering process rather than a novel discovery, the pipeline is clear and informative. The bottleneck, such as the token length gap between image and audio, is intriguing and has been partly resolved in this paper.

**Strengths:**

1. This paper systematically analyzes the limitations of current MLLM models and focuses on their critical aspect: any-to-any understanding and generation capabilities.
2. This paper examines images, videos, and speech to explore how to create a generalized model for all modalities while noting differences among them, such as variations in token length distribution.
3. This paper compares MIO with various existing MLLM models across different tasks to showcase its effectiveness.

**Weaknesses:**

1. This paper uses speech-enhanced pretraining to address variations in token length distribution. However, this approach is more of a patch than a well solution.
2. This paper states that "GPT-4o has showcased... However, it is closed-source." Therefore, will MIO be made publicly available? We do not require an additional closed-source copy of GPT-4o for research purposes.
3. This paper aims to address any-to-any understanding and generation. It should possess capabilities beyond existing methods. For example, can this model generate a talking-hand video in a few-shot setting that encompasses text understanding, as well as video and speech generation?

**Questions:**

See the weaknesses.

---

> ### Author Response · Authors · 2024-11-30
> **Responses for Q1**
>
> Thanks for your kind feedback. Here are some clarifications:
>
> Q1: speech-enhanced pre-training stage is not a solution but a patch.
>
> A1: The **speech-enhanced pretraining** stage is designed to address the challenge posed by imbalanced token amount across different modalities, specifically for speech data, by increasing the speech data ratio during the whole training process. We can call our strategy as “staircase-style speech ratio warmup” since the speech data ratio is increased stage by stage. This approach ensures that the model can consume the speech data more effectively without compromising the performance of the other modalities, such as image-text alignment.
>
> To validate the effectiveness of the three-stage approach, we compare stage 1, stage 2, and stage 3, with each stage being independent and from scratch with the corresponding data mixing ratio (note that “stage 2” is different from “stage 1+2”). After training for 1K to 2K steps, the experimental results are shown below:
>
> | Setup | Data Composition | COCO Captioning (CIDEr) | COCO Imagen (human evaluation) | LibriSpeech ASR (WER) |
> | --- | --- | --- | --- | --- |
> | stage 1 (1K) | 12:2:0:2 | 10.84 | vaguely relevant to the caption and with a clear outline of the described object. | 90.19 |
> | stage 2 (1K) | 2:2:6:6 | 4.21 | low-quality images, irrelevant to the caption. | 89.90 |
> | stage 3 (1K) | 2:1:1:12 | 2.80 | low-quality images, irrelevant to the caption. | 85.32 |
> | stage 1 (2K) | 12:2:0:2 | 25.03 | vaguely relevant to the caption and with a clear outline of the described object. | 90.03 |
> | stage 2 (2K) | 2:2:6:6 | 4.53 | low-quality images, irrelevant to the caption. | 88.02 |
> | stage 3 (2K) | 2:1:1:12 | 2.86 | low-quality images, irrelevant to the caption. | 80.73 |
>
> Note that the data composition is in format of “(image-text pair data) : (language-only data) : (image-text interleaved data + video data) : (speech-text pair data)”
>
> We can observe that the image-text alignment by “stage 2”-only improves much slower than by “stage 1”. And there is a risk for very early performance saturation for image-text alignment when directly using the data mixture of stage 2. Moreover, since the image-text paired corpus is much larger and higher-quality than the image-text interleaved corpus, to ensure the pretraining can consume most of the image-text paired data, we exclude the image-text interleaved data in the first stage. And to further unlock the models’ abilities for image-text interleaved generation, we add these data back into the pretraining once the training loss of the stage 1 nearly converges. As for the third stage — speech enhancement pretraining, since the speech tokens are much more enormous, we maximize its data ratio to ensure most of the speech data is consumed once the performance for other modalities during stage 2 is saturated.
>
> In conclusion, such staircase-style warmup is designed to avoid early performance saturation for image-text representation alignment and to ensure that most of the training data for each modality (esp., speech) is trained. Therefore, our staged stragety is not only a patch, but also a solution.

---

> ### Author Response · Authors · 2024-11-30
> **Responses for Q2**
>
> Q2: not open-source
>
> A2: We have open-sourced both the models and codes, but we cannot provide the links due to the anonymity requirements. You can refer to the attached MIO supplemental materials\MIO_anonymous for the anonymous codes. Unfortunately, we cannot upload our models either in the supplemental materials or in the anonymous github repository due to the size limit.

---

> ### Author Response · Authors · 2024-11-30
> **Responses for Q3**
>
> Q3: capabilities beyond existing methods.
>
> A3: We have shown the advanced abilities of our model, such as visual storytelling, chain of visual thought, visual guideline generation, instructional image editing, etc., in Figure 4. Moreover, our model also supports the trimodal understanding of image+text+speech, as illustrated in Table 8.
>
> ---
>
> Thanks again for your valuable comments. Hope our clarifications can address most of your concerns, and we are looking forward to further feedback.

---

> ### Author Response · Authors · 2024-12-02
> **Looking forward to further feedback**
>
> Hi, Reviewer dfgg,
>
> Thank you for your insightful comments. With the discussion period nearing its end, we would be grateful if you could let us know whether our responses have sufficiently addressed your concerns.

---

> > ### Comment · Reviewer_dfgg · 2024-12-03
> >
> > This project is interesting, and the main concern regarding open-source has been addressed. I will keep the score for acceptance.

---

### Official Review · Reviewer_MGdu · 2024-11-04

**Soundness:** 2
**Presentation:** 3
**Contribution:** 2
**Rating:** 5
**Confidence:** 4

**Summary:**

This paper proposes a discrete token-based foundation model to unify understanding and generation capability in four modalities: speech, text, images, and videos with a proposed four-stage training recipe.

**Strengths:**

- The paper explores the potential of unifying four different modalities in a DIDO manner using existing tokenizers within a single causal MLLM.
- It also enables the generation of multimodal output in interleaved sequences.

**Weaknesses:**

- The paper demonstrates limited novelty in comparison to existing work within the research community.
- The overall performance of the proposed model is less competitive and lacks comprehensiveness. For example:
  - In the image understanding task, the dual-modal baselines used for comparison are relatively weak, with some considered obsolete.
  - In the image generation task, the reliance on the CLIP score metric is limiting, as it primarily focuses on text alignment and overlooks important aspects like structural integrity and aesthetics. Additionally, it has been noted to be less aligned with human preferences [1-3].
- There is a lack of both qualitative and quantitative ablation or analysis to justify key model design choices, such as the tokenizer selection, data composition, and training setup.

[1] Kirstain, Yuval, et al. "Pick-a-pic: An open dataset of user preferences for text-to-image generation." Advances in Neural Information Processing Systems 36 (2023): 36652-36663.

[2] Xu, Jiazheng, et al. "Imagereward: Learning and evaluating human preferences for text-to-image generation." Advances in Neural Information Processing Systems 36 (2024).

[3] Wu, Xiaoshi, et al. "Human preference score: Better aligning text-to-image models with human preference." Proceedings of the IEEE/CVF International Conference on Computer Vision. 2023.

**Questions:**

Please refer to the weaknesses.

---

> ### Author Response · Authors · 2024-11-30
> **Responses for Q1 - Q2**
>
> Q1: limited novelty in comparison to existing work.
>
> A1: While MIO builds upon prior research, we believe it makes meaningful contributions to the field of multimodal foundation models. Specifically, we list the novel features of our work in Table 1.
>
> | Models        | I/O Consistency | Uni. Bi. SFT | Multi-Task SFT | Speech I/O  | Video I/O   | Voice Output | MM. Inter. Output | Modeling  |
> |---------------|-----------------|--------------|----------------|-------------|-------------|--------------|--------------------|-----------|
> | Emu1          | ✗               | ✗            | ✓              | ✗/✗         | ✓/✓         | ✗            | ✗                  | CICO      |
> | Emu2          | ✓               | ✗            | ✓              | ✗/✗         | ✓/✓         | ✗            | ✗                  | CICO      |
> | SEED-LLaMA    | ✓               | ✓            | ✓              | ✗/✗         | ✓/✓         | ✗            | ✓                  | DIDO      |
> | AnyGPT        | ✓               | ✓            | ✗              | ✓/✓         | ✗/✗         | ✗            | ✗                  | DIDO      |
> | CM3Leon       | ✓               | ✓            | ✓              | ✗/✗         | ✗/✗         | ✗            | ✗                  | DIDO      |
> | Gemini        | ✗               | ✓            | ✓              | ✓/✗         | ✓/✗         | ✗            | ✗                  | CIDO      |
> | Transfusion   | ✗               | ✗            | ✗              | ✗           | ✗           | ✗            | ✗                  | AR+Diff  |
> | **MIO (Ours)**| ✓               | ✓            | ✓              | ✓/✓         | ✓/✓         | ✓            | ✓                  | DIDO      |
>
> We highlight the following novel aspects:
>
> 1. **Support for Multimodal Interleaved Sequence Generation**:
>
>     MIO uniquely supports the generation of multimodal interleaved sequences, a capability absent in prior works such as Emu and CM3Leon. This feature enables advanced tasks, including interleaved video-text generation, chain-of-visual-thought reasoning, and instructional image editing, which we demonstrate in Figure 2.
>
> 2. **Addressing Limitations in Existing Models**:
>     - Unlike SEED-LLaMA, Emu1, and Emu2, MIO integrates speech input and output.
>     - MIO also surpasses the scope of prior works like AnyGPT, which lack advanced capabilities such as voice generation, interleaved video-text generation, etc.
> 3. **Emergent Abilities and Generalist Features**:
>
>     As a model that combines four modalities with both understanding and generation capabilities, MIO demonstrates emergent abilities that arise from its any-to-any foundation. Examples include chain-of-visual-thought reasoning and visual guideline generation (Figure 2), showcasing the model's unique strengths.
>
>
> By addressing these gaps and extending the capabilities of multimodal foundation models, we believe MIO makes a meaningful contribution to the field.
>
> ---
>
> Q2: limited performance and outdated baselines for image understanding.
>
> A2: We add two strong baselines, i.e., Qwen-VL-Chat (13B)[1] and LLaVA 1.5 (7B)[2], in the Table 3:
>
> | **Models** | Imagen | Speech | COCO | VQAv2 | OKVQA | VizWiz | SEED Bench |
> | --- | --- | --- | --- | --- | --- | --- | --- |
> | Emu-Base (14B) | ✔️ | ❌ | 112.4 | 52.0 | 38.2 | 34.2 | 47.3 |
> | Emu-I (14B) | ❌ | ❌ | 120.4 | 57.2 | 43.4 | 32.2 | 58.0 |
> | SEED-LLaMA-I (8B) | ✔️ | ❌ | 124.5 | 66.2 | 45.9 | 55.1 | 51.5 |
> | AnyGPT (8B) | ✔️ | ✔️ | 107.5 | - | - | - | - |
> | Flamingo (9B) | ❌ | ❌ | 79.4 | 51.8 | 44.7 | 28.8 | 42.7 |
> | Flamingo (80B) | ❌ | ❌ | 84.3 | 56.3 | 31.6 |  | - |
> | Kosmos-1 (1.6B) | ❌ | ❌ | 84.7 | 51.0 | - | 29.2 | - |
> | MetaLM (1.7B) | ❌ | ❌ | 82.2 | 41.1 | 11.4 | - | - |
> | IDEFICS-I (80B) | ❌ | ❌ | 117.2 | 37.4 | 36.9 | 26.2 | 53.2 |
> | CM3Leon (7B) | ✔️ | ❌ | 61.6 | 47.6 | 23.8 | 37.6 | - |
> | InstructBLIP (8.1B) | ❌ | ❌ | - | - | - | 34.5 | 58.8 |
> | Qwen-VL-Chat (13B) | ❌ | ❌ | - | 78.2 | 56.6 | 38.9 | 58.2 |
> | LLaVA 1.5 (7B) | ❌ | ❌ | - | 78.5 | - | 50.0 | 58.6 |
> | **MIO-Instruct (7B)** | ✔️ | ✔️ | 120.4 | 65.5 | 39.9 | 53.5 | 54.4 |
>
> We can observe that MIO-Instruct still demonstrates minor performance gaps despite less image resolution (Qwen-VL-Chat is trained on 448x448 images, LLaVA 1.5 is trained on 336x336 images, while ours is trained on 224x224 images) and the additional support for speeches and image generation.
>
> [1] Qwen-vl: A frontier large vision-language model with versatile abilities
>
> [2] Improved Baselines with Visual Instruction Tuning

---

> ### Author Response · Authors · 2024-11-30
> **Responses for Q3**
>
> Q3: limited metrics for image generation (structural integrity, aesthetics, human preferences)
>
> A3: We compute two additional automatic metrics, i.e., SSIM and Aesthetic Predictor v2.5 for the evaluation of structural integrity and aesthetics, respectively. SSIM (Structural Similarity Index Measure) evaluates the perceptual similarity between the generated images and the ground-truth images, focusing on luminance, contrast, and structure, with scores ranging from -1 (dissimilar) to 1 (identical). Aesthetic Predictor V2.5 is a SigLIP-based predictor that evaluates the aesthetics of an image on a scale from 1 to 10 (10 is the best).
>
> In addition, we randomly select 100 image descriptions from MS-COCO test set, and use each model to generate images accordingly for human preference evaluation. We ask 3 annotators to rank 3 images generated by the 3 models: “given the image description, which image is preferred?” The average ranking of MIO’s, AnyGPT’s, and Emu’s generated images are 1.2 (MIO), 2.9 (AnyGPT), 1.9 (Emu). MIO aligns the best with the human preference. The percentage agreement between the three annotators (calculated as the number of cases with identical rankings by all annotators divided by 100) is 82.3%, indicating a high consistency in the human evaluation.
>
> The results are as follows:
>
> | Dataset | MS-COCO | MS-COCO | Flickr30K | Flickr30K | MS-COCO subset |
> | --- | --- | --- | --- | --- | --- |
> | Metric | SSIM | Aesthetic Predictor v2.5 | SSIM | Aesthetic Predictor v2.5 | Human Avg. Ranking |
> | Emu | 0.1749 | 3.733 | 0.1451 | 3.893 | 1.9 |
> | AnyGPT | 0.1960 | 3.954 | 0.1585 | 4.251 | 2.9 |
> | MIO | 0.2307 | 4.019 | 0.1727 | 4.326 | 1.2 |
>
> Accordingly, MIO achieves the best performance in terms of both structural integrity and aesthetics. Moreover, we recognized the importance of introducing human evaluation for image generation. As shown by the metric of human averaged rankings, we observe that MIO is most preferred by the real users compared with AnyGPT and Emu. We promise to add the references regarding the human preference for image generation (i.e., [1-3]) in our revised version.
>
> [1] Kirstain, Yuval, et al. "Pick-a-pic: An open dataset of user preferences for text-to-image generation." Advances in Neural Information Processing Systems 36 (2023): 36652-36663.
>
> [2] Xu, Jiazheng, et al. "Imagereward: Learning and evaluating human preferences for text-to-image generation." Advances in Neural Information Processing Systems 36 (2024).
>
> [3] Wu, Xiaoshi, et al. "Human preference score: Better aligning text-to-image models with human preference." Proceedings of the IEEE/CVF International Conference on Computer Vision. 2023.

---

> ### Author Response · Authors · 2024-11-30
> **Responses for Q4**
>
> Q4: lack of qualitative and quantitative ablations or analysis for tokenizer selection, data composition, and training setup.
>
> A4: For the image tokenizer selection, we compare three tokenizers qualitatively in Figure 3. We can observe that the other two tokenizers demonstrate too low quality to enable us to leverage comparable quantitative evaluation. For the speech tokenizer, we directly leverage the SpeechTokenizer since it has more token efficiency (200 tokens/seconds) compared with SoundStream[1] (600 tokens/seconds) and HiFiCodec[2] (400 tokens/seconds). Some more efficient speech tokenizers such as WavTokenizer[3] and GLM-4-Voice-Tokenizer[4], are our concurrent works.
>
> For the data composition and the training setup, since investigating numerous data compositions is unaffordable due to high training costs, between each stage (stage1 → stage2, stage2 → stage3), we simply maximize the micro batch size for the data type of interest (i.e., image-text paired data for stage1, image-text interleaved data and video data for stage2, and speech data for stage3) while keep the ratio for other data types reasonable (i.e., mostof the training corpora for each modality should be trained on). We compare stage 1, stage 2, and stage 3, with each stage being independent and from scratch with the corresponding data mixing ratio (note that “stage 2” is different from “stage 1+2”). After training for 1K to 2K steps, the experimental results are shown below:
>
> | Setup | Data Composition | COCO Captioning (CIDEr) | COCO Imagen (human evaluation) | LibriSpeech ASR (WER) |
> | --- | --- | --- | --- | --- |
> | stage 1 (1K) | 12:2:0:2 | 10.84 | vaguely relevant to the caption and with a clear outline of the described object. | 90.19 |
> | stage 2 (1K) | 2:2:6:6 | 4.21 | low-quality images, irrelevant to the caption. | 89.90 |
> | stage 3 (1K) | 2:1:1:12 | 2.80 | low-quality images, irrelevant to the caption. | 85.32 |
> | stage 1 (2K) | 12:2:0:2 | 25.03 | vaguely relevant to the caption and with a clear outline of the described object. | 90.03 |
> | stage 2 (2K) | 2:2:6:6 | 4.53 | low-quality images, irrelevant to the caption. | 88.02 |
> | stage 3 (2K) | 2:1:1:12 | 2.86 | low-quality images, irrelevant to the caption. | 80.73 |
>
> Note that the data composition is in format of “(image-text pair data) : (language-only data) : (image-text interleaved data + video data) : (speech-text pair data)”
>
> We can observe that the image-text alignment by “stage 2”-only improves much slower than by “stage 1”. And there is a risk for very early performance saturation for image-text alignment when directly using the data mixture of stage 2. Moreover, since the image-text paired corpus is much larger and higher-quality than the image-text interleaved corpus, to ensure the pretraining can consume most of the image-text paired data, we exclude the image-text interleaved data in the first stage. And to further unlock the models’ abilities for image-text interleaved generation, we add these data back into the pretraining once the training loss of the stage 1 nearly converges. As for the third stage — speech enhancement pretraining, since the speech tokens are much more enormous, we maximize its data ratio to ensure most of the speech data is consumed once the performance for other modalities during stage 2 is saturated.
>
> [1] Soundstream: An end-to-end neural audio codec
>
> [2] HiFi-Codec: Group-residual Vector quantization for High Fidelity Audio Codec
>
> [3] WavTokenizer: An Efficient Acoustic Discrete Codec Tokenizer for Audio Language Modeling
>
> [4] GLM-4-Voice-Tokenizer: https://github.com/THUDM/GLM-4-Voice
>
> ---
>
> Thanks again for your kind reviews! We are looking forward to your further feedback.

---

> ### Author Response · Authors · 2024-12-02
> **Looking forward to further feedback**
>
> Hi, Reviewer MGdu,
>
> Thank you for your valuable comments. As the discussion period is coming to a close, we would appreciate it if you could let us know whether our responses have addressed your concerns.

---

### Author Response · Authors · 2024-12-03
**Seeking further feedback**

Dear Reviewers and Area Chairs,

We sincerely appreciate the valuable feedback and constructive suggestions provided by the reviewers. In response, we have carefully addressed the comments, offering detailed clarifications and additional experimental results.

As the discussion phase is nearing its conclusion (**less than 10 hours remain**), we kindly request any further feedback you might have at your earliest convenience.

Thank you for your time and consideration.

Sincerely,
MIO Authors

---

### Author Response · Authors · 2024-12-04
**Summary of the Rebuttal**

Dear Reviewers and Chairs,

We sincerely thank all the reviewers for their insightful and constructive feedback. We appreciate that you recognized the significance of our work in advancing multimodal foundation models. We have carefully considered all comments and have made revisions to strengthen our paper.

Our key contributions are summarized as follows and listed in Table 1:

1. **Support for Multimodal Interleaved Sequence Generation**: MIO uniquely enables the generation of multimodal interleaved sequences, a capability absent in prior works. This feature allows for advanced tasks such as interleaved video-text generation, chain-of-visual-thought reasoning, instructional image editing, etc.
2. **Addressing Limitations in Existing Models**: MIO surpasses the scopes of prior models by supporting both understanding and generation across four modalities in a unified manner.
3. **Emergent Abilities and Generalist Features**: Combining multiple modalities with both understanding and generation capabilities, MIO demonstrates emergent abilities arising from its any-to-any foundation.

Following the insightful suggestions of the reviewers, we have made the following revisions:

- **Reviewer MGdu:**
    - **Added Strong Baselines**: We included two strong baselines, Qwen-VL-Chat (13B) and LLaVA 1.5 (7B), in our image understanding experiments (Table 3) to address concerns about outdated baselines.
    - **Expanded Metrics for Image Generation**: To address limitations in evaluation metrics for image generation, we added SSIM and Aesthetic Predictor v2.5 scores and conducted human evaluations to assess structural integrity, aesthetics, and human preferences (Table 12).
    - **Provided Qualitative and Quantitative Ablations**: We added analyses on tokenizer selection, data composition, and training setup to justify key design choices (A4 for Reviewer MGdu).
- **Reviewer Uas7**:
    - **Evaluated on Text-Only Benchmarks**: We assessed our models on text-related benchmarks like MMLU to demonstrate MIO's language capabilities (Table 7).
    - **Ablations and Loss Curves**: We provided ablation experiments on the three-stage pretraining strategy and discussed the similarities between our strategy and one-stage pretraining with speech ratio warmup strategy (A3.1 for Reviewer Uas7). We also provided the loss curves for the three pretraining stages (Figure 3).
    - **Evaluated Speech Dialogue Performance**: We conducted evaluations on speech-to-speech tasks and compared our model with concurrent works like LLaMA-Omni (Table 13).
    - **Enhanced TTS Evaluation**: We added more metrics and benchmarks for TTS evaluation, including speaker similarity, WER, and human evaluation, on GLOBE, LibriSpeech, and VCTK benchmarks, comparing MIO with state-of-the-art speech models and AnyGPT (Table 5, Table 14, and A6 for Reviewer Uas7).
    - **Provided Audio Demos**: We included non-cherry-picked audio demonstrations in the supplementary materials to enhance understanding of the model's capabilities.
- **Reviewer dfgg**:
    - **Clarified the Three-Stage Pretraining Strategy**: We provided detailed explanations and experimental results to demonstrate that the speech-enhanced pretraining stage is an effective solution, not just a patch (A1 for Reviewer dfgg).
    - **Open-Sourced the Model**: We have open-sourced both the models and codes (anonymized for review), addressing concerns about accessibility for research purposes.
    - **Highlighted Advanced Capabilities**: We included demonstrations and explanations of MIO's capabilities beyond existing methods, such as advanced multimodal tasks (Figure 4).
- **Reviewer GVGf**:
    - **Provided Qualitative Comparisons**: We added qualitative comparisons between MIO and Emu for image generation in the supplementary materials to illustrate the advantages of our model.
    - **Clarified Multimodal Inference**: We explained how the model outputs fixed-length image, speech, and video tokens during inference without special decoding constraints (A2 for Reviewer GVGf).
    - **Analyzed Model Performance**: We provided insights into why MIO outperforms other models in image understanding and generation, discussing the benefits of the DIDO approach over CICO (A3 for Reviewer GVGf).
    - **Added More Baselines**: We included results for Emu2 in our experiments for a more comprehensive comparison (A6 for Reviewer GVGf).
    - **Emphasized the Importance of Video Keyframe Generation over Full Video Generation and Discussed Differences Between MIO and NeXT-GPT** (A4 and A5).

We are glad that our responses addressed most of your concerns (Reviewer Uas7 and Reviewer dfgg), even though the scores weren't raised. We still look forward to further feedback from Reviewer GVGf and Reviewer MGdu (by editing the initial reviews or scores).

Thank you all once again for your valuable feedback and engagement. Your comments have significantly helped us improve our paper.

Best regards,

MIO Authors

---

### Meta-Review · Area_Chair_F4PH · 2024-12-21

**Metareview:**

This paper introduces MIO, a new multimodal foundation model designed for understanding and generating content across speech, text, images, and videos. MIO uses a discrete token-based approach (Discrete-In-Discrete-Out or DIDO). The model employees a multi-stage training recipe from different pre-trainings to sft on tasks from different modalities to ensure the robust cross-modality performance.

The reviewers agrees this is an important direction of investigating the unified LLM for any-to-any understanding and generation capabilities. The paper provides a good systematical analysis of the limitations of the current approach. The proposed multi-stage training is also reasonable. Reviewers raised two major concern. The first is the novelty is limited heavily relays on the existing work and mostly connecting them together, and second the performance is less competitive such as vision understanding tasks and TTS tasks.

**Additional Comments On Reviewer Discussion:**

The authors tried to address the major concerns aforementioned above, but the reviewers were not convinced with the newly added experiments.

---

### Decision · Program_Chairs · 2025-01-22

Reject